# Structure of the homodimeric androgen receptor ligand-binding domain

Marta Nadal[1,2,3,*], Stefan Prekovic[4,*], Nerea Gallastegui[1,2,*], Christine Helsen[4], Montserrat Abella[1,2], Karolina Zielinska[1], Marina Gay[5], Marta Vilaseca[5], Marta Taulès[6], Adriaan B. Houtsmuller[7,8], Martin E. van Royen[7,8], Frank Claessens[4], Pablo Fuentes-Prior[2,3,**] & Eva Estébanez-Perpiñá[1,2,**]

The androgen receptor (AR) plays a crucial role in normal physiology, development and metabolism as well as in the aetiology and treatment of diverse pathologies such as androgen insensitivity syndromes (AIS), male infertility and prostate cancer (PCa). Here we show that dimerization of AR ligand-binding domain (LBD) is induced by receptor agonists but not by antagonists. The 2.15-Å crystal structure of homodimeric, agonist- and coactivator peptide-bound AR-LBD unveils a 1,000-$Å^2$ large dimerization surface, which harbours over 40 previously unexplained AIS- and PCa-associated point mutations. An AIS mutation in the self-association interface (P767A) disrupts dimer formation *in vivo*, and has a detrimental effect on the transactivating properties of full-length AR, despite retained hormone-binding capacity. The conservation of essential residues suggests that the unveiled dimerization mechanism might be shared by other nuclear receptors. Our work defines AR-LBD homodimerization as an essential step in the proper functioning of this important transcription factor.

[1] Department of Biochemistry and Molecular Biomedicine, Institute of Biomedicine (IBUB) of the University of Barcelona (UB), Barcelona 08028, Spain. [2] Bases Estructurals de Processos Fisiopatològics Fonamentals, 2014-SGR-01214, Agència de Gestió d'Ajuts Universitaris i de Recerca (AGAUR), Barcelona 08010, Spain. [3] Molecular Bases of Disease, Biomedical Research Institute Sant Pau (IIB Sant Pau), Barcelona 08025, Spain. [4] Molecular Endocrinology Laboratory, KU Leuven, Herestraat 49, Leuven 3000, Belgium. [5] Mass Spectrometry Core Facility, Institute for Research in Biomedicine (IRB Barcelona), The Barcelona Institute of Science and Technology (BIST), Barcelona 08028, Spain. [6] Unitat de Citometria, Centres Científics I Tecnològics (CCIT), Universitat de Barcelona (UB), Barcelona 08028, Spain. [7] Department of Pathology, Erasmus MC, Wytemaweg 80, Rotterdam 3015 CN, The Netherlands. [8] Erasmus Optical Imaging Centre, Erasmus MC, Wytemaweg 80, Rotterdam 3015 CN, The Netherlands. * These authors contributed equally to this work. ** These authors jointly supervised this work. Correspondence and requests for materials should be addressed to P.F.-P. (email: pfuentes@santpau.cat) or to E.E.-P. (email: evaestebanez@ub.edu).

The androgen receptor (AR/NR3C4) belongs to the steroid receptor subfamily of nuclear receptors (NRs), which also includes the glucocorticoid receptor (GR/NR3C1), mineralocorticoid receptor (MR/NR3C2), progesterone receptor (PR/NR3C3) and oestrogen receptors α and β (ERα/NR3A1; ERβ/NR3A2). Steroid receptors are major therapeutic targets, due to their pivotal role in a number of endocrine-related diseases[1,2]. The AR, in particular, is critically important for normal development and homeostasis of male and female reproductive organs and their physiology[3]. To date, more than a thousand cases with pathogenic mutations affecting the human *AR* gene have been reported[4]. These variations can generate a dysfunctional receptor and lead to androgen insensitivity syndromes (AIS)[5], which depending on the clinical phenotype are classified as complete (CAIS), partial (PAIS) or mild (MAIS). On the other hand, a large number of gain-of-function *AR* mutations have been associated with castration-resistant prostate cancer (PCa), one of the leading causes of cancer death in men worldwide[6,7]. The rich clinical information on AR-related pathologies continues to provide a detailed knowledge on the structure-function relationships for this transcription factor, as well as for the other NRs.

Structurally, the AR is similar to other NRs consisting of an N-terminal domain, followed by an almost strictly conserved DNA-binding domain (DBD), an interdomain linker or hinge, and a C-terminal ligand-binding domain (LBD). The LBD contains the internal ligand-binding pocket (LBP) and two major solvent-exposed surfaces responsible for interaction with coregulators, activation function 2 (AF-2) and binding function 3 (BF-3)[8–11] and Supplementary References. Structural information has been gathered on several full-length NRs either by detailed X-ray crystallography, or through small-angle X-ray scattering and electron microscopy at lower resolution[12–16]. These biophysical investigations have revealed conflicting data that does not allow a unified paradigm of full-length NR architecture at present time[17,18]. Regarding the AR, there is no experimental structural information accounting for the multi-domain receptor. Thus far, a structure of the DNA-bound AR-DBD dimer[19], as well as numerous structures of monomeric AR-LBD complexed with agonists or antagonists have been elucidated[20–23]. This structural information on isolated domains and consideration of reported mutations have guided our previous modelling attempts of the full-length protein[24]. Since the individual AR domains have autonomous functions (nuclear translocation, coactivator recruitment, DNA and ligand binding), several intra- and inter-domain interactions are essential for the integration of input and output signals required for proper AR functioning. Establishing the order of key events leading to gene activation and the molecular basis of allosteric control of the various AR functions still remains a major challenge[25,26]. In this regard, the identification of the dimerization mechanisms and their physiological relevance may profoundly impact the development of new AR therapeutics.

Here we present the crystal structure of the human AR-LBD homodimer bound to its natural agonist, dihydrotestosterone (DHT) and provide in addition strong evidence for its crucial role in receptor functioning. Most importantly, over forty published *AR* mutations linked to AIS or PCa have been found to cluster at this interface providing significant *in vivo* support for the current homodimeric AR-LBD structure. Disease-associated mutations were found to affect the dimer interface and lead to functional dysregulation of key AR actions, corroborating the physiological significance of this protein–protein interaction site.

## Results

**The LBD of AR interacts with UBA3.** Ubiquitin-activating enzyme 3 (UBA3) was identified in yeast two-hybrid screens to bind to DHT-bound AR-LBD, used as bait against human adult brain and prostate cDNA libraries. UBA3 has previously been shown to interact directly with ERα (ref. 27). The androgen-dependent UBA3 interaction with the AR relies on the presence of an LxxLL NR-interacting motif. A synthetic UBA3 peptide comprising this motif, S$^{59}$TESLQFLLDTCKV$^{72}$ (S$^{59}$-V$^{72}$), was found by surface plasmon resonance (SPR) to bind with high affinity to liganded AR-LBD ($K_D = 30.6 \pm 0.7$ nM; Fig. 1a). Crystals of AR-LBD grown in the presence of this peptide diffracted X-rays up to a resolution of 2.15 Å, which allowed solution and refinement of the structure of the complex (Fig. 1b). (See Table 1 for data collection and refinement statistics, as well as structure quality parameters). Electron densities in and around coactivator binding AF-2 grooves could be safely interpreted as corresponding to residues S$^{62}$-T$^{69}$ of the UBA3 peptide. These residues adopt an α-helical conformation with the side chains of Leu residues L$^{63}$ and L$^{67}$ inserted into the AF-2 pockets of the AR-LBD (Fig. 1b), similar to the structures documented before for other LxxLL peptide motifs[23].

**The crystal structure of the AR-LBD homodimer.** All AR-LBD structures deposited in the Protein Data Bank (PDB) to date belong to the same crystal form (orthorhombic space group P2$_1$2$_1$2$_1$), and feature a monomer in the asymmetric unit (ASU)[20–23] and Supplementary References. In contrast, the current AR-LBD crystal structure belongs to the monoclinic space group (C2) and presents four independent, helically arranged LBD molecules in the ASU (Fig. 1e,f; details of the final electron density map are shown in Fig. 1c,d). Two of these LBD monomers form a symmetrical 'core dimer' upon burial of ≈ 1,000 Å$^2$ of solvent-exposed surface from each molecule (Figs 1e,f and 2a–e), while two peripheral AR-LBDs associate more loosely to the BF-3 grooves of each of these monomers (Figs 1d,f and 2f,h).

All four molecules in the ASU can be superimposed on the previously solved monomeric structures, indicating an essential conservation of the LBD scaffold (r.m.s.d. of 0.56 Å when compared with PDB entry 1T7T). Significant structural differences were limited to the more N-terminal residues (E669-F674) and to some loops that were mobile or even partially disordered in most monomeric AR-LBD structures. (Numbering refers to the recently revised sequence of full-length human AR). This is the case of L1–3 (E682-S697), but in particular of the basic L9–10 (C845-N849), which is clearly defined by electron density in the current structure (Fig. 2c; Supplementary Fig. 1a). The stabilizing structural effect of inter-LBD contacts is also reflected by the lower temperature factors of the current structure compared with those of monomeric AR-LBD refined at a similar resolution (Supplementary Fig. 1b). Due to their potential physiological relevance, inter-monomer contacts will be briefly described below.

Core dimer: the two monomers in the AR-LBD core dimer are arranged 'head-to-head' around a local pseudo twofold axis with both AF-2 pockets facing opposite directions and separated by over 60 Å (Fig. 2a,b). In essence, if the 'left' AR-LBD is displayed in the standard orientation (that is, with helix H1 and the AF-2 groove facing the viewer), the 'right' AR-LBD shows its 'back' surface (H10-H11). The protein–protein interface is centred on residues from helix H5 and the L5-S1 loops of both partners, with additional contributions made by residues from H1 and H7-H9, L1–3, and β-strand S1 (Fig. 2a). The

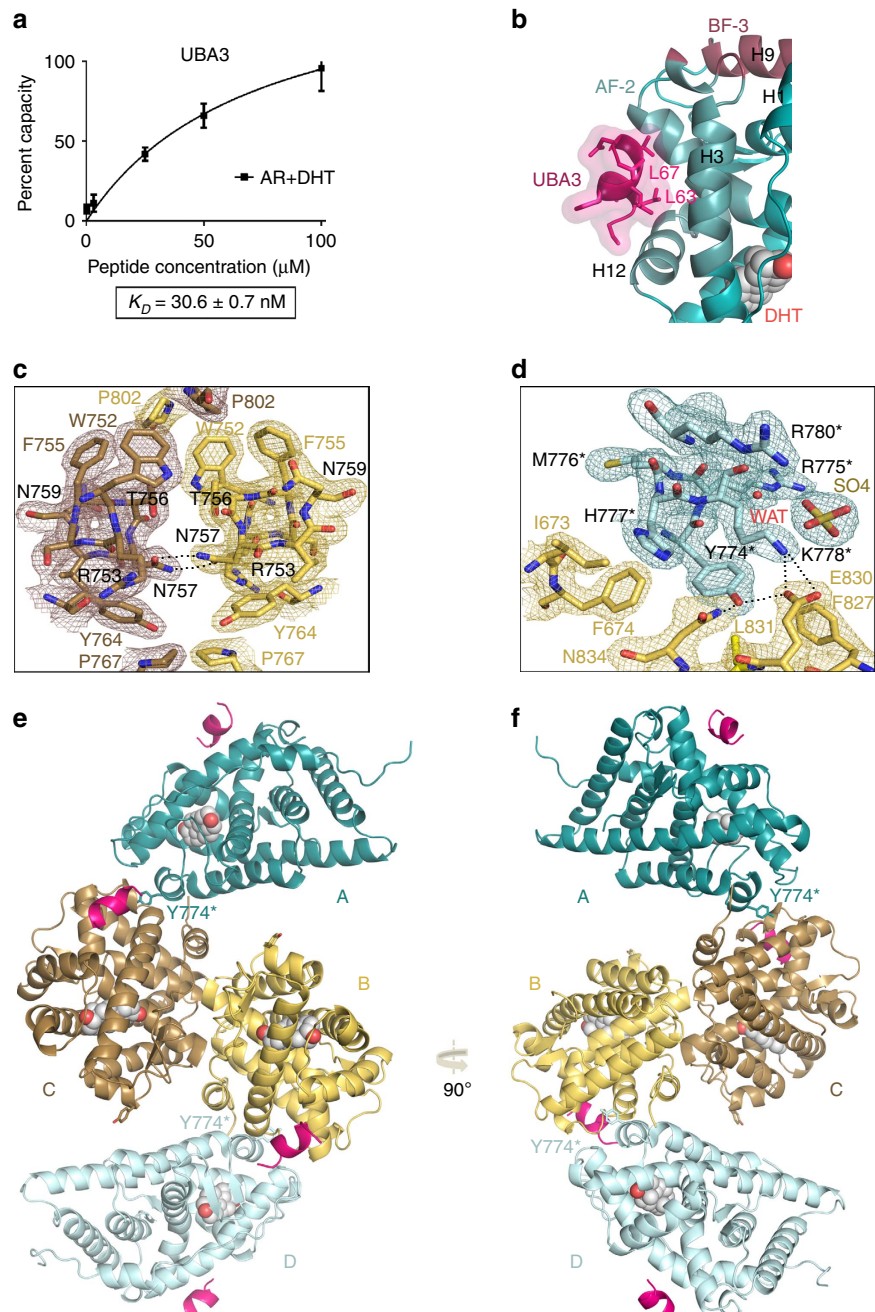

**Figure 1 | Crystal structure of AR-LBD in complex with UBA3 peptide.** (**a**) An UBA3 peptide comprising the canonical LxxLL motif binds tightly to AR-LBD. The results of SPR studies conducted in triplicate are shown. (**b**) Closeup around the AF-2 binding groove with the bound UBA3 peptide shown as a cartoon (pink, with leucine side chains represented as sticks). AR is also depicted as a cartoon with AF-2 and BF-3 binding areas highlighted in brighter blue and magenta, respectively, and the bound DHT moiety in sphere representation. (**c,d**) Details of the final electron density map. Most relevant AR-LBD residues are represented as sticks and H-bonds with black dotted lines. (**c**) Closeup showing major interactions across the interface of the core dimer composed by the arbitrarily labelled molecules B (in yellow) and C (in brown). Electron density is shown as either a brown or yellow mesh contoured at 1σ. (**d**) Closeup showing docking of H6 from peripheral AR-LBD molecule A (pale blue) into the BF-3 pocket of AR-LBD molecule B (yellow). Residues from the peripheral monomer are marked with an asterisk. (**e,f**) Two views of the AR-LBD crystal structure with the four independent AR-LBD molecules (**a–d**) found in the ASU. Notice that AR-LBD monomers B (yellow) and C (brown) form a symmetrical core dimer, while the two peripheral AR-LBD labeled as (**a**) shown (teal) and (**d**) (pale blue) are associated to the BF-3 grooves of (**b,c**) respectively.

two AR-LBDs are tilted by ∼20° perpendicular to the pseudo twofold axis relating the partners (Fig. 2b). This tilting results in a slightly asymmetric dimer structure, which alleviates the electrostatic repulsion of the basic L9–10, but in particular of acidic patches centred on residues D691 (L1–3) and D768 (LS1-S2) from both monomers.

Residues from the two monomers are arranged symmetrically along the pseudo twofold axis, although some side chain conformations and therefore the details of intermolecular contacts differ slightly. At the core of the dimer interface, both P802 residues are nested in aromatic cages formed by the side chains of V685, W752, F755 and Y764 from a neighbouring

**Table 1 | Crystallographic data and refinement statistics.**

| PDB | 5JJM | | |
|---|---|---|---|
| *CRYSTAL PARAMETERS* | | | |
| Space group | C2 (monoclinic) | | |
| Molecules/asymmetric unit | 4 | | |
| *Cell dimensions* | | | |
| a, b, c (Å) | 91.09, 90.83, 157.23 | | |
| β (°) | 90.07 | | |
| Matthews coefficient | 2.71 | | |
| Solvent content (%) | 54.61 | | |
| *DATA COLLECTION* | | | |
| Wavelength (Å) | 0.9 | | |
| | Overall – Inner Shell – Outer Shell | | |
| Low resolution limit (Å) | 78.61 | 78.61 | 2.27 |
| High resolution limit (Å) | 2.15 | 6.80 | 2.15 |
| $R_{merge}$ | 0.073 | 0.042 | 0.456 |
| Total number of observations | 214,583 | 6,866 | 31,533 |
| Total number of unique | 69,531 | 2,294 | 10,052 |
| Multiplicity | 3.1 | 3.0 | 3.1 |
| Completeness (%) | 99.6 | 99.8 | 99.2 |
| I/σ(I) | 6.8 | 11.8 | 1.4 |
| Mean I/σ | 8.9 | 19.7 | 2.5 |
| *REFINEMENT* | | | |
| *Fit to data used in refinement* | | | |
| Resolution range (Å) | 157.23-2.15 | | |
| Reflections used refinement | 66,522 | | |
| Completeness (%) | 99.77 | | |
| *Final model* | | | |
| Number of non-hydrogen atoms | 8,848 | | |
| Number of solvent molecules | 120 | | |
| *R factors* | | | |
| R value (working+test set) | 0.203 | | |
| R value (working set) | 0.201 | | |
| Free R value | 0.243 | | |
| Free R value test set size (%) | 5.0 | | |
| Free R value test set count | 3,444 | | |
| *Fit in the highest resolution bin* | | | |
| Bin resolution range high (Å) | 2.147 | | |
| Bin resolution range low (Å) | 2.203 | | |
| Reflections in bin (working set) | 4,706 | | |
| Bin completeness (working+test set) | 96.91 | | |
| Bin R value (working set) | 0.233 | | |
| Free R value test set count | 254 | | |
| Bin Free R value | 0.289 | | |
| Average B factors (overall, Å$^2$) | 40.54 | | |
| *R.m.s deviations* | | | |
| Bond lengths (Å) | 0.026 | | |
| Bond angles (°) | 1.97 | | |
| *MODEL QUALITY* | | | |
| Favoured rotamers | 88.3% | | |
| Ramachandran plot most favoured (%) | 97 | | |
| Ramachandran plot allowed (%) | 3 | | |
| MolProbity, clash score all atoms | 3.62 (99[th] percentile)* | | |
| MolProbity score | 1.89 (88[th] percentile)* | | |

*100th percentile is the best among structures of comparable resolution; 0th percentile is the worst.

monomer (Fig. 2c–e and Supplementary Fig. 2e). Noteworthy, π-stacking interactions of residues W752 and F755 rigidify the H5 helices, which appears to be essential for this H5-H5′ interface. Particularly strong van der Waals (vdW) interactions are formed between the F755-P802 pairs (Figs 1c and 2d). Further, residue V685 rests against the Y764 phenolic group from the neighbouring monomer, and also engages in additional vdW contacts with V758 (Fig. 2d). Other residues symmetrically opposed upon dimerization are P767, as well as the polar residues T756 and N757, which allows formation of hydrogen (H-) bonds across the dimer axis (Figs 1c and 2c,d). The interface is further strengthened by H-bonds between the guanidinium group of R761 and the main chain carbonyl oxygen atoms of E679, A680 and/or E682 from the neighbour

LBD (Fig. 2e). Most importantly, R753 interacts with both interface residues such as N757 and with the bound hormone by means of its side chain (Fig. 2c–e; see also below).

BF-3-mediated contacts: the BF-3 pockets of the core dimer partners harbour the short H6 from neighbouring molecules (Figs 1d and 2f–h), which exclude ≈ 370/385 Å$^2$ of solvent-exposed surface. The hot spot residue at this interface is Y774* whose aromatic side chain inserts between those of F674 and F827. Binding is strengthened by important vdW contacts of the Y774* side chain with P724/G725 and L831 (Figs 1d–f and 2f–h). In addition, Y774* donates a H-bond to E830, which also forms a salt bridge with K778* (Fig. 1d). Finally, residues H777* and Y782* additionally contribute to anchor H6 in BF-3 (Fig. 2f).

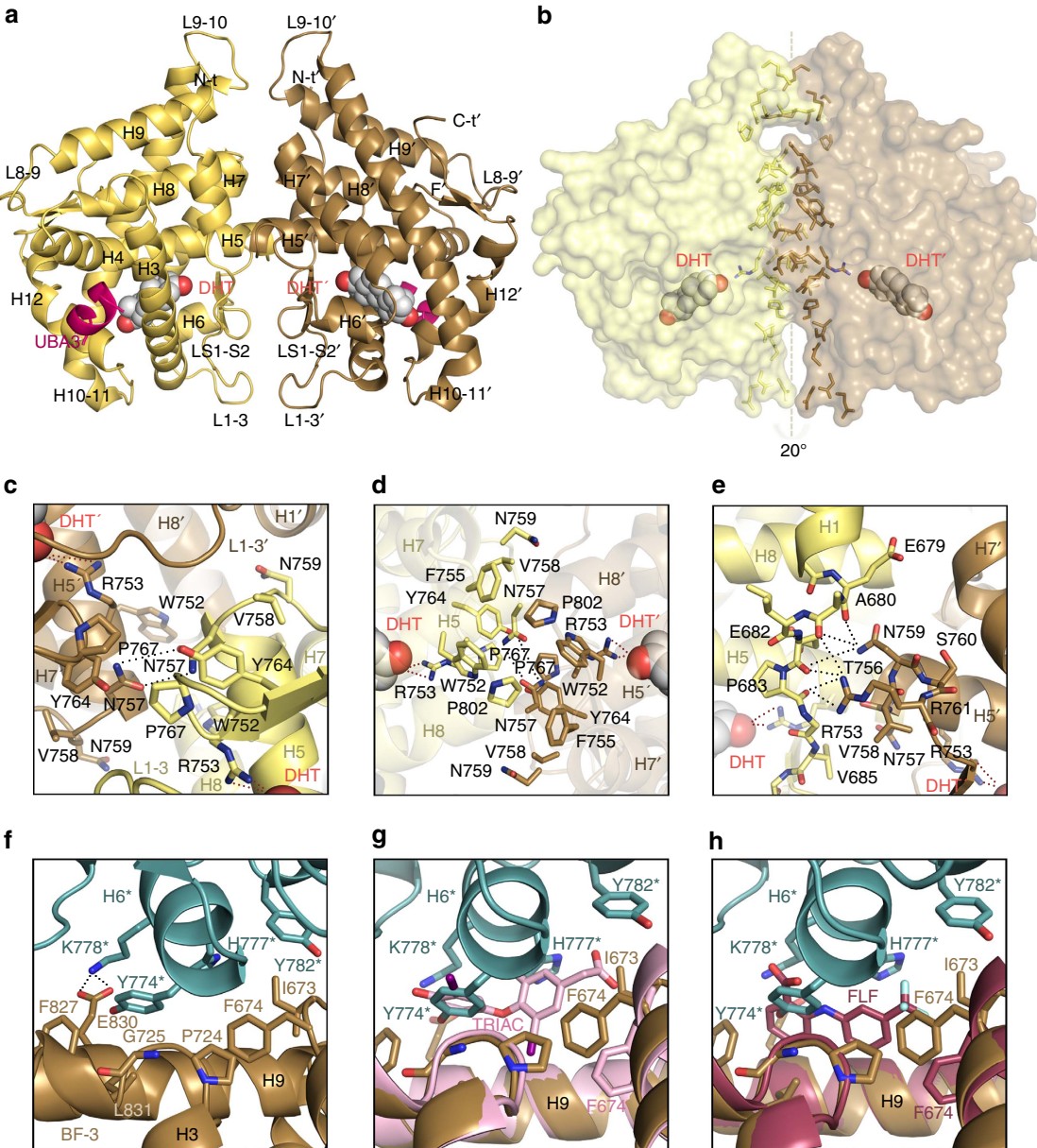

**Figure 2 | Details of the AR-LBD dimer interface.** (**a**) Overall structure of the AR-LBD core dimer. The two monomers are depicted as cartoons, with monomer B (yellow) in standard orientation and monomer C in brown; helices and loops are marked. The hormone (dihydrotestosterone, DHT) and the UBA3 peptide are shown as spheres and as a cartoon, respectively. (**b**) Surface representation of the AR-LBD homodimer shown in the same orientation and coloured yellow and brown as in **a**. The side chains of residues involved in direct inter-monomer contacts are represented as sticks, coloured according to the monomer they belong to. The DHT moieties are depicted as color-coded spheres (oxygen, red; carbon, yellow or brown). The 'right' AR-LBD monomer is titled by ~20° perpendicular to the pseudo twofold axis relating the partners, which results in a slightly asymmetric dimer. (**c–e**) Closeups of the AR-LBD dimer interface highlighting major inter-domain contacts. Residues are shown as color-coded sticks (oxygen, red; nitrogen, blue; carbon, yellow or brown) and labelled. Hydrogen bonding interactions are indicated with black dots. (**f**) Closeup of the H6 helix from monomer A docking onto the BF-3 pocket of monomer B. Relevant residues are depicted as sticks and H-bonds as black dotted lines. The Tyr774* residues of the peripheral monomers occupy topologically equivalent positions as the outer ring of TRIAC (**g**) or the benzoic ring of FLF (**h**). Residues from the peripheral monomers are marked with an asterisk. See also Supplementary Fig. 1.

**AR-LBD homodimerizes through the H5-mediated interface.** To confirm AR-LBD dimerization in solution, we first assessed the capacity of DHT-liganded AR-LBD for non-covalent 1:1 self-association by SPR. Analysis of the kinetics of self-interaction revealed rapid association ($k_a = (8.1 \pm 0.2) \times 10^3 \, \mathrm{M}^{-1}\mathrm{s}^{-1}$) and dissociation phases ($k_d = 0.072 \pm 0.002 \, \mathrm{s}^{-1}$), from which an affinity constant ($K_D$) = $8.8 \pm 0.08 \, \mu\mathrm{M}$ could be calculated (Fig. 3a). Analysis of the affinity of protein–protein interactions yielded a similar $K_D$ of $1.90 \pm 0.05 \, \mu\mathrm{M}$ (Fig. 3b). These $K_D$ values

are consistent with SPR results previously reported for other NR LBDs[28,29]. Preliminary cross-linking experiments with glutaraldehyde revealed formation of AR-LBD dimers, in addition to higher-order species (Supplementary Fig. 2a), and prompted us to analyse in more detail the dimerization process in solution.

To demonstrate that AR-LBD homotypic interactions in solution involve the same surfaces identified in the current crystal structure, we took advantage of the presence of

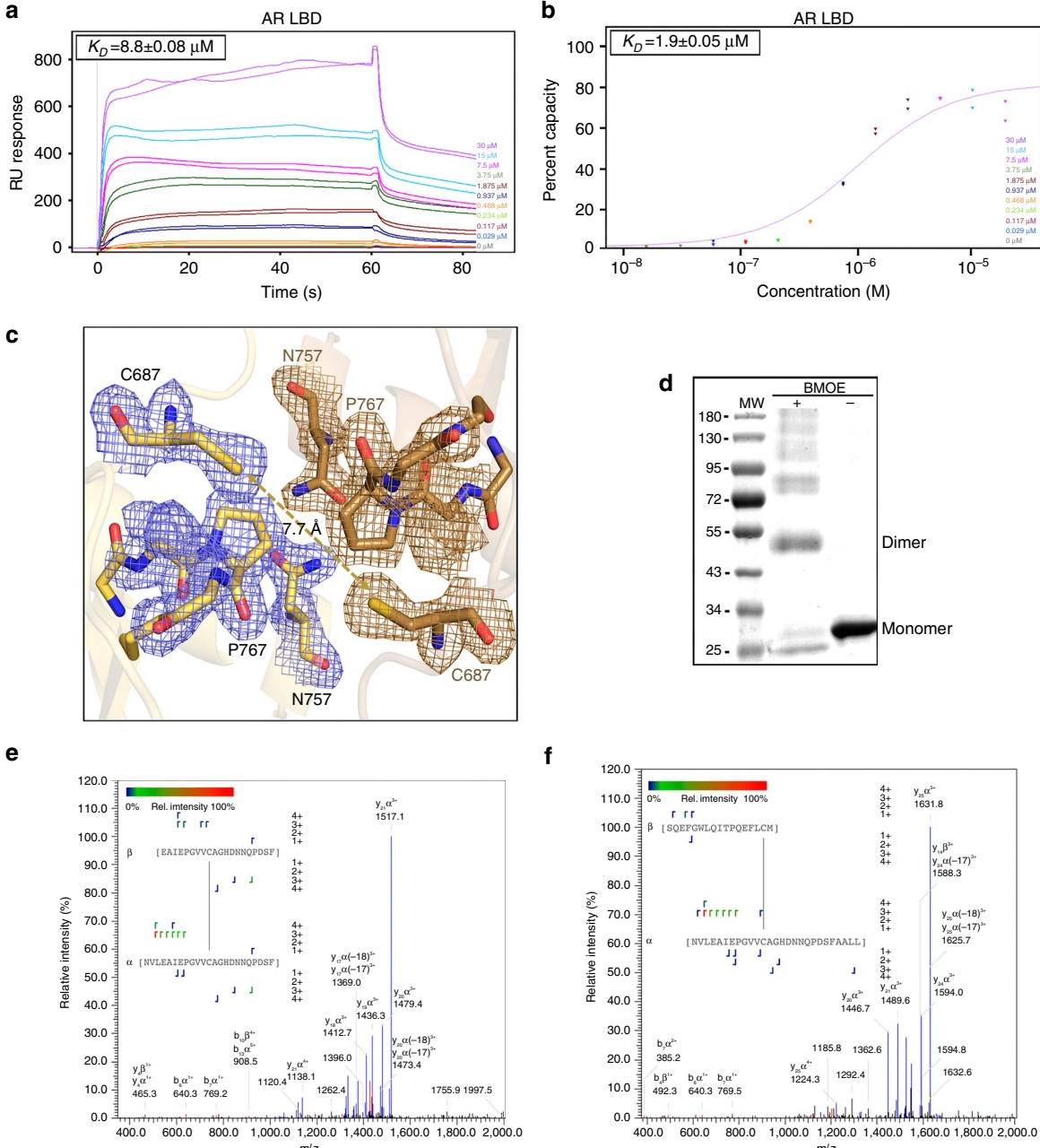

**Figure 3 | AR dimerizes in solution through the H5–H5′ interface.** SPR analysis of AR-LBD self-association by kinetics (**a**) or affinity (**b**). The results of experiments conducted in duplicate are shown along with the respective calculated affinity constants. (**c**) Closeup of the core dimer interface highlighting the close proximity between the C687 Sγ atoms from both monomers. (**d**) BMOE-induced cross-linking of AR-LBD. The molecular masses (in kDa) of standard proteins are shown at the left side of the gel (MW). Notice detection of an AR-LBD dimer along with bands corresponding to higher-order aggregates in the presence but not in the absence of the crosslinker. (**e**,**f**) Representative MS/MS spectra identifying BMOE-crosslinked peptides that include residues C687 from both monomers. See also Supplementary Fig. 2 and Supplementary Table 1.

unique pairs of residues at each of the interfaces. First, we noticed that four out of the six cysteine residues in the AR-LBD are solvent accessible (C670, C687, C845 and C853), and thus capable of reacting with sulfhydryl-reactive small molecules. Further, inspection of the homodimer structure immediately reveals that only residues C687 from the two core monomers are located close enough to be simultaneously engaged by the short-arm crosslinker, bis-maleimidoethane (BMOE; Fig. 3c). As expected from these observations, incubation of AR-LBD in the presence of BMOE resulted in rapid and almost quantitative formation of a covalent dimer (Fig. 3d). This is in addition

to intramolecular bridges between residue C670 and either C853 or C845, which are detectable as a more rapidly migrating band corresponding to monomeric AR-LBD in Fig. 3d. To verify that residues C687 are indeed responsible for BMOE-mediated dimer formation, we analysed by mass spectrometry chymotryptic digests of monomeric and dimeric AR-LBD. As expected, various combinations of BMOE-crosslinked peptides N676–F698, E679–F698 and N676–L702 were detected only in dimeric AR-LBD, and their identity verified by MS/MS (see Supplementary Table 1 for a summary of detected peptides and Fig. 3e,f for representative MS/MS spectra).

In addition, we assessed the possible relevance of the H6-BF-3 interaction in solution by taking advantage of the presence of a salt bridge between residues K778* and E830 (see Fig. 2f and above). Incubation of purified AR-LBD with the zero-length crosslinker, 1-ethyl-3-(3-dimethylaminopropyl)carbodiimide (EDC) revealed indeed appearance of a faint band corresponding to the dimer, but most of the material remained as a monomer (Supplementary Fig. 2b). We conclude that the preferred conformation of homodimeric AR-LBD in solution is centred on the much larger, symmetric H5-H5′ interface.

**Dimerization regulation by ligands and a CAIS mutation.** The experiments described above were conducted using highly purified, isolated AR-LBD proteins. To study AR-LBD dimerization in living cells, we performed acceptor-bleaching fluorescence resonance energy transfer microscopy (FRET) experiments[30]. To this end, AR-LBD constructs were genetically fused with either enhanced yellow fluorescent protein (EYFP) at the N-terminus or with ECFP at the C-terminus (Fig. 4a) and co-expressed in Hep3B cells, essentially as previously described[31]. As expected from the inclusion of the R630-K634 NLS in these constructs, the fusion proteins were localized in the nucleus, also in the absence of hormone (Fig. 4b). As illustrated in Fig. 4b, no FRET signal was detected in the absence of added hormone, but it was strongly induced by AR agonists (testosterone, dihydrotesterone and R1881). By contrast, no FRET effect was observed in the presence of the AR antagonists enzalutamide (Enza), bicalutamide (Bic) or hydroxyflutamide (OHF). We conclude that AR-LBD homodimerization follows hormone occupation of the LBP, and that current antiandrogens function, at least partly, by blocking this hormone-induced event.

To verify the relevance of interface residues for receptor dimerization in vivo, we introduced either a mutation predicted to favour homodimer formation (Y764C, identified in both PCa and AIS patients) or the CAIS-associated mutation, P767A, in both EYFP- and ECFP-tagged AR-LBD fusions (Fig. 4c). In line with the solvent-exposed position of the exchanged residues in monomeric LBD, both mutant proteins were correctly folded, as indicated by their retained ligand-binding properties (Fig. 4d,e). As expected from our in silico predictions (Supplementary Fig. 3), the PCa mutation Y764C mutation allowed ligand-induced dimerization as evidenced by a DHT-induced FRET signal (Fig. 4c). Importantly, when the CAIS P767A mutation was introduced in both EYFP and ECFP AR-LBD fusions, the hormone was no longer able to induce a FRET signal (Fig. 4c). We conclude that AR-LBD dimerization is controlled by the ligand and that point mutations of interface residues interfere with receptor homodimerization without affecting ligand binding.

The dimer interface is critical for full-length AR activity. To prove the functional relevance of the dimer interface for full-length AR functioning we analysed the effect of selected mutations found either in AIS (W752R and P767A) or both AIS and PCa (Y764C), or that represent more drastic replacements of naturally occurring variants (V758K, R761E and R856E). With exception of the Y764C exchange, all mutations were anticipated to have a negative effect on homodimer stability.

The selected mutations had varying effects on the activity of the NR when tested on a classical reporter construct (Fig. 5a). Variants AR(P767A) and AR(R856E) were virtually inactive in this transactivation assay, while W752R displayed a ten-fold reduced maximal activity. AR(V758K) showed a lower maximal activity as well as a more than 10-fold reduction in $EC_{50}$, while for R761E the response was reduced two-fold. Interestingly, the maximal response of the Y764C variant previously shown

to retain the homodimerization ability almost doubled that of wild-type AR. These dramatic effects on transactivation activity caused by mutations that affect the dimer interface were not due to large differences in ligand affinity and binding capacity, as evaluated in whole-cell binding assays for the full-length AR variants W752R, Y764C and P767A (Fig. 5b,c). Indeed, the binding capacity for 1 nM mibolerone and the relative affinity for DHT were reduced two-fold for W752R and P767A, and remained unaffected for AR(Y764C). Furthermore, the different AR constructs were expressed to similar levels, as demonstrated by immunoblotting (Fig. 5d,e and Supplementary Fig. 6). We conclude that AR-LBD dimerization via helix H5 and nearby areas (Supplementary Table 2) is critical for the transcriptional activity of the AR.

**Discussion**

The contribution of LBD dimerization to the physiological activity of the AR has remained controversial for a long time. Here we present the crystal structure of an AR-LBD homodimer, along with biochemical and functional evidence of its relevance in vivo. First, we notice that the large inter-monomer interface ($\approx 1,000\,\text{Å}^2$) compares well with other solved structures of NR homo- and heterodimers, and is significantly larger than the previously reported GR-LBD and PR-LBD homodimers[32–35]; see also Supplementary Figs 2 and 5 and below. Our SPR results provide direct evidence for AR-LBD self-association in solution, and we could unambiguously demonstrate that the H5-centred dimerization mode revealed in the crystal structure is also preferred in solution. Strongly supporting the relevance of this arrangement in vivo, the isolated AR-LBD was shown by FRET to dimerize also in a more complex, cellular environment. Importantly, the control of dimerization by AR agonists and antagonists correlates with the well-known effect of these compounds on the activity of the full-length receptor. Finally, the CAIS-associated mutation, P767A, which is predicted to impair dimerization because of its negative impact on vdW interactions, disrupts DHT-induced homodimer formation.

The role of the AR-LBD interface in the functioning of the full-size receptor was further verified through functional analysis of carefully selected mutations of residues exposed on the contact surface. Replacements were predicted to either disrupt (W752R, V758K, R761E, P767A and R856E) or enhance dimerization (Y764C) without major impact on the 3D structure of the AR-LBD and without altering other important functions (Supplementary Fig. 3, Supplementary Notes). In fact, we detected only minor changes in ligand-binding properties for AR variants W752R and P767A. By contrast, all substitutions tested had dramatic negative consequences on receptor functioning in our assay: W752R, P767A and R856E nearly completely disrupted activity, while V758K and R761E strongly reduced transactivation. Our results are consistent with published data on variants P767A (ref. 36), R856C/H (refs 37,38) and V758K/I/A (refs 39,40). While these mutations were previously reported to affect transactivation and/or ligand binding in different assays, our current structure now points out their detrimental effects on homodimer formation as the primary cause of receptor malfunctioning (Supplementary Fig. 3).

Interestingly, the Y764C variant showed a higher transactivation potential than the wild-type receptor. A similar higher activity has been reported for another mutation, T756A, which would also stabilize the dimer according to our analyses[39]. How dimer stabilization enhances transactivation is unclear at the moment. Possibly it affects one or more downstream AR functions, but the direct consequences of dimer formation on the biological activities of the AR need further investigation.

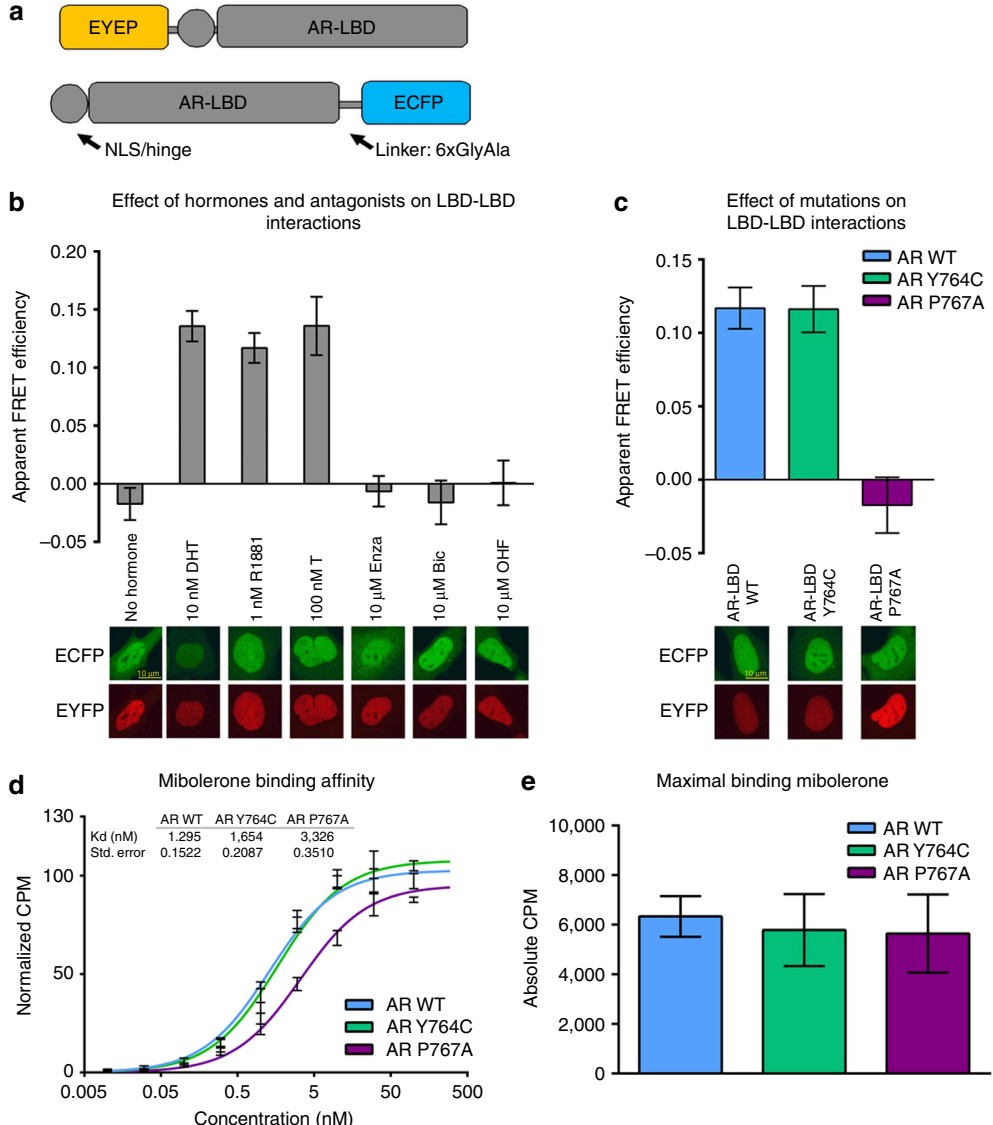

**Figure 4 | Functional characterization of homotypic AR-LBD interactions by FRET.** (**a**) Schematic representation of the generated fusion proteins. (**b**) Acceptor photobleaching FRET of N-terminal and C-terminal fusions of AR-LBD shows agonist-induced interactions (DHT, ($n = 65$), T ($n = 32$), and R1881 ($n = 48$), while no interactions were observed without hormone ($n = 44$) or when antagonists (Bic ($n = 38$), Enza ($n = 46$), and OHF ($n = 44$)) were bound to the LBD (mean values and standard error of the mean of at indicated number of cells are shown). Representative confocal images of cells expressing the fusions of AR with EYFP/ECFP in the presence of these compounds are displayed below the bars. (**c**) Acceptor photobleaching FRET of indicated proteins shows loss of interaction for the AR P767A mutant ($n = 67$) when compared with the WT ($n = 59$), but not for the Y764C mutant ($n = 63$; mean values and s.e.m. of indicated number of cells are shown). Representative confocal images of Hep3B cells transiently expressing the indicated protein in the presence of DHT are displayed below the bars. (**d**) Binding affinity of the EYFP-AR-LBD fusion protein for the AR agonist mibolerone. (**e**) Maximal binding of WT and mutant AR for mibolerone (mean values and standard error of the mean of three experiments with three technical replicates each are shown).

Altogether, the current findings demonstrate that the LBD dimerization surface is critical for the transcriptional activity of the AR and for androgen physiology. There is now evidence for inter- and intramolecular interactions at the levels of the N/C interactions[41], at the level of the DBD[19], and of the LBD (this work). While for the first two interactions the spatial-temporal distributions have been determined[26], we now need to integrate LBD dimerization in the chronology of gene activation by the AR. The functional implications of disrupting or stabilizing the dimerization process are further illustrated by significant correlations between naturally occurring AR-LBD mutations and important pathologies, as discussed below.

To date, almost 200 point mutations in the AR-LBD have been linked to either AIS and/or PCa. Previously available structures of monomeric AR-LBD allowed for a straightforward rationalization of the impact of pathogenic mutations that directly affect hormone binding, as 22 of them (17% of all reported point mutations) map to the LBP. Mutations in residues that line AF-2 and BF-3 grooves explain further 28 (22%) and 21 (16%) variants, respectively[20–23]. However, many other residues mutated in AIS or PCa are well exposed on the surface of AR-LBD monomer, and are unlikely to directly affect protein structure or coregulator binding. The current crystal structure offers a likely molecular explanation for over 40 *AR* mutations that affect residues buried in the AR-LBD dimer

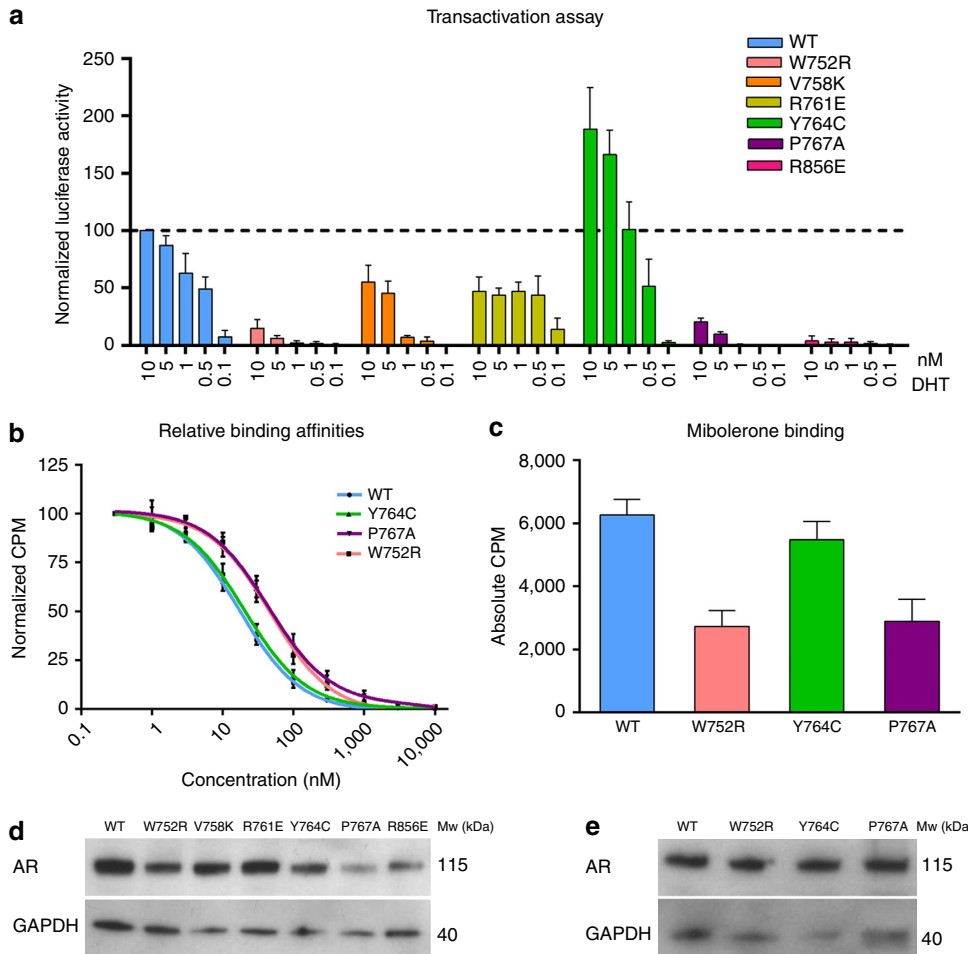

**Figure 5 | Functional validation of the AR-LBD dimer interface.** To investigate the impact of mutations predicted to influence dimerization of the full-length receptor, luciferase reporter and whole-cell competition assays were performed in PC-3 and COS-7 cells, respectively. The mean and s.e.m. of four independent experiments with three technical replicates each are shown for both assays. (**a**) Transactivation assays were performed with increasing concentrations of DHT (from 0.1 to 10 nM). (**b**,**c**) Determination of relative binding affinities for DHT and of maximum binding of mibolerone. (**d**,**e**) Western blot analysis of wild-type and mutant AR variants from the experiments shown in panels (**a**–**c**), respectively.

interface (33%; Fig. 6). This is in particular the case for recurrent mutations of residues F755, N757, V758, N759, R761 and P767. This large mutational 'hot spot' across the LBD strongly suggests a functional relevance of the AR-LBD homodimer.

We used various bioinformatics tools for a systematic in-depth rationalization of the impact of disease-linked point mutations on AR protein folding as well as on homodimer formation and/or stability (Supplementary Methods, Supplementary Fig. 3, Supplementary Tables 3 and 5 and Supplementary Notes). Interestingly, AIS-associated mutations are spread all over the interface (Fig. 6a), while PCa mutations mostly cluster at the core of the dimer (Fig. 6b). This might reflect the selection of mutations during AR targeting therapies, which is different from the more random distribution of mutations seen in AIS. The major implications of the current structure for disease-associated point mutations are summarized in Supplementary Results.

While the physiological relevance of the core LBD dimer for AR functioning seems to be solidly demonstrated by a wealth of structural and functional data, including a large number of naturally occurring point mutations, there are certain more speculative issues raised by the current structure that we would like to address below.

Signal transduction in NRs at the molecular level is mediated by long-range communication between topographically distinct (non-overlapping) binding sites (e.g., LBP, AF-2, and BF-3 in the LBD, as well as the DBD). These allosteric transitions may involve subtle, reversible conformational changes that are still under intense investigation. How allosteric effects are propagated across the different intra- or inter-functional surfaces (N-terminal domain, DBD, hinge and LBD) are still not fully elucidated[8,42–45]. The current structure of the AR homodimer highlights an additional level of communication between the main functional sites of the AR-LBD partners (inter-domain allostery; Fig. 6a,b). The strictly conserved residue R753 is the central element in coupling the dimerization partners as it makes crucial direct contacts with either agonists or antagonists in the LBP, while simultaneously contributing to LBD dimer assembly (Figs 1c; 2c,e and 6c)[20–23]. The intricate residue network in this area thus directly connects hormone binding with dimer formation. This is corroborated by the FRET data, which show that agonist binding induces dimerization (Fig. 4c). The P767A mutation impairs vdW interactions across the dimer interface (Fig. 2d), thus partially disrupting this network, reducing dimerization, and thus ultimately inactivating the AR. Of note, AR(P767A) retains ligand-binding capability, albeit with a lower affinity (Fig. 4d).

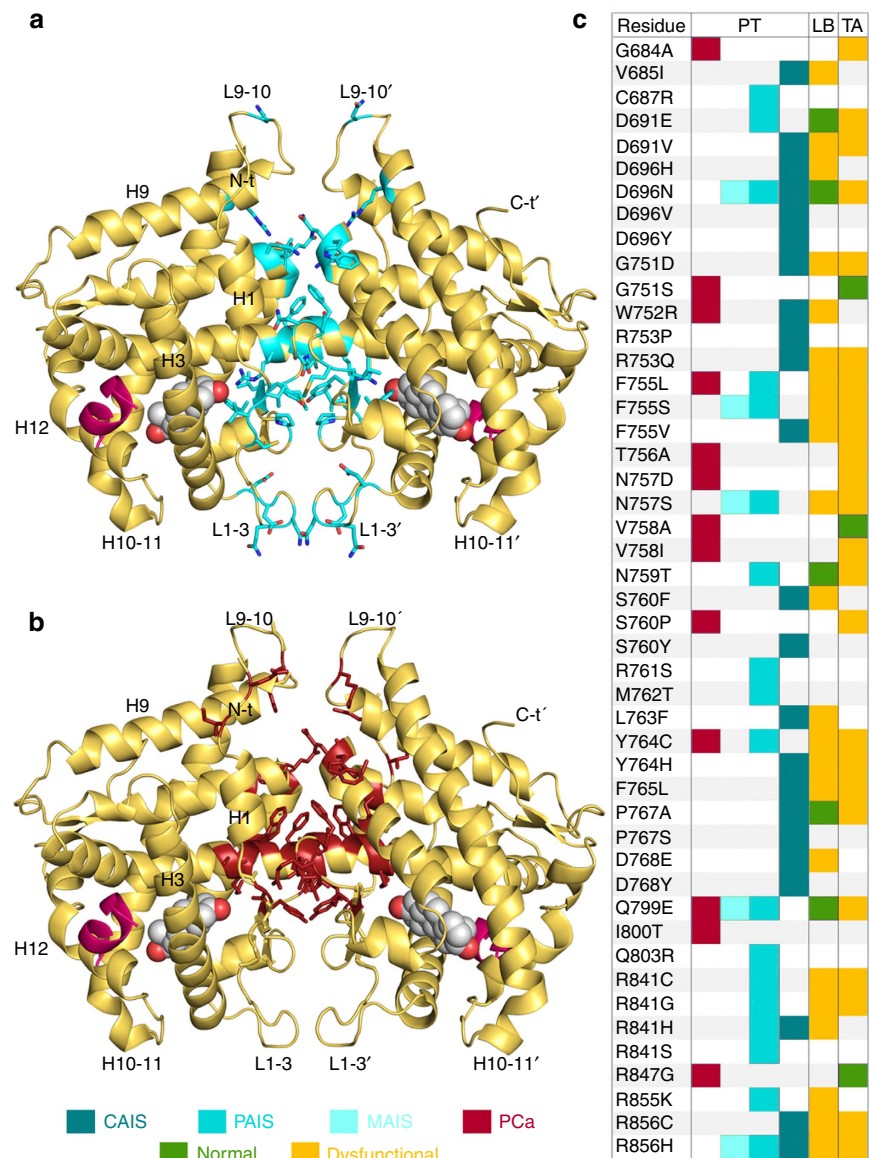

**Figure 6 | Mutations associated with AIS and PCa cluster in the AR-LBD dimer interface.** Cartoon representation of the AR-LBD dimer (yellow) with the side chains of all mutated interface residues shown with all their non-hydrogen atoms as sticks, coloured blue for AIS (**a**) or red for PCa (**b**). (**c**) A complete list of *AR* missense mutations that affect interface residues reported to date, along with their associated phenotypes. Mutations have been taken from the Androgen Receptor Gene Mutations Database (http://androgendb.mcgill.ca/). For a detailed bioinformatics analysis of the predicted impact of these exchanges see Supplementary Fig. 3, Supplementary Notes and Supplementary Tables 3 and 5.

This indicates that ligand binding can occur independently from dimerization.

At the same time, LBP occupancy in one LBD may influence the ligand-binding capacity of the second monomer through the H5-H5′ interface. Whether this allosteric effect fully explains the reduction in ligand affinity of the P767A variant remains to be investigated. Along these lines, it is noteworthy that binding of antagonist R-bicalutamide destabilizes helix H5 in monomeric LBD, as indicated by significantly higher B factors[22] PDB 1Z95; see also Supplementary Fig. 1b). Thus, it would seem that interference with LBD dimerization is a major action mechanism of AR antagonists (Fig. 4b).

The LBP-to-LBP′ allosteric connection could be expected to synchronize the AF-2 and BF-3 grooves from the two interacting partners (Fig. 7b), in line with proposals that binding interactors or mutations at remote sites lead to functional changes at (an)other area(s) either through alteration of receptor shape

and/or its dynamics[8,42,43,45–48]. In particular, dimerization might directly influence ligand binding and/or remodel the AF-2 landscape through long-range allosteric communication facilitating or disrupting productive protein–protein interactions with key coregulators. Of note, the AF-2 pockets of both partners in the AR dimer remain accessible for interactions with coregulatory complexes, in alignment with currently accepted models of full-length NR functioning derived from EM data[16]. Elucidating the allosteric pathways communicating across the dimer interface and in particular with the AF-2 and BF-3 interacting surfaces of the dimer partners needs further investigation as it has been studied in other NRs[44]. Furthermore, we postulate that LBD dimerization could also influence activities of other AR domains. This is corroborated by the retained ligand-binding ability of the transcriptionally inactive, dimer-disrupting AR mutants, W752R and P767A (Fig. 5). At a higher level, it is interesting to note that allosteric linkages

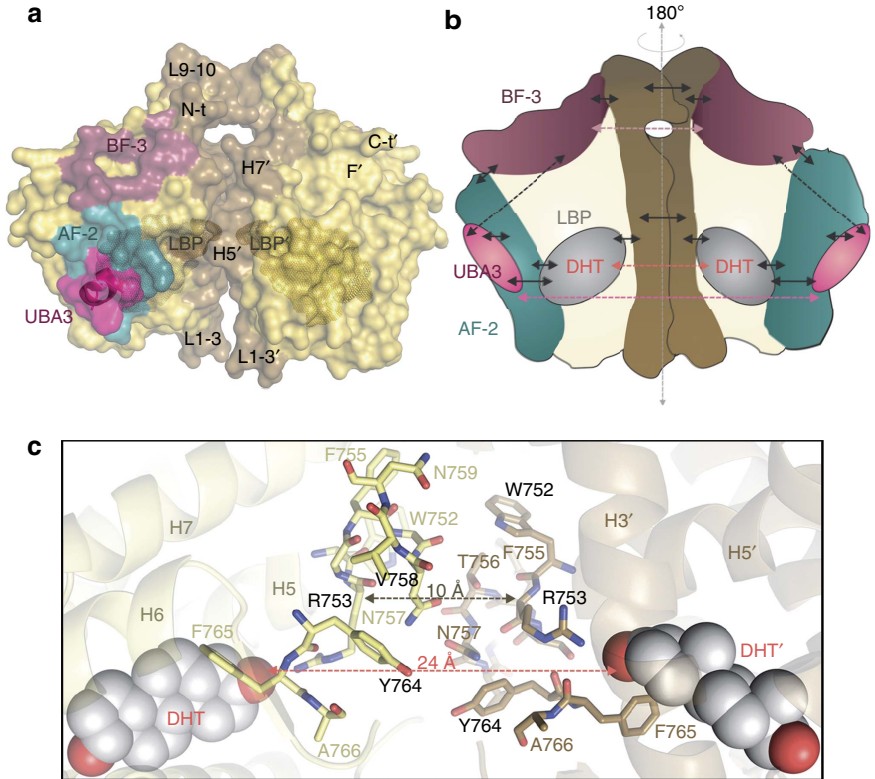

**Figure 7 | Proposed allosteric communication pathways across the AR-LBD dimer interface.** (**a**) Surface representation of the AR homodimer. The dimer interface (brown), the AF-2 groove (blue) and the BF-3 pocket (raspberry) are highlighted. Residues that form or line the LBP are shown with a Connolly dot surface and the UBA3 peptide as a pink surface. (**b**) Schematic representation of the proposed intra- and inter-domain allosteric pathways in AR-LBD. Solid arrows indicate short-range communication networks, while dashed arrows point to long-range interactions. (**c**) Close-up of the dimer interface highlighting allosteric communication between the LBPs across the dimer interface. The distances between the two R753 residues and the two DHT moieties are given.

between DNA and ligand binding and transactivation have been proposed earlier for the AR[24].

The LBDs of the subfamily of oxosteroid NRs (AR, GR, MR and PR) differ in several key structural features from the ER subclass and the RARα-RXRα heterodimers[32,49] (Supplementary Figs 4 and 5). In particular, oxosteroid NRs lack a hydrophobic motif at the N-terminal end of H10, which is an essential element of the canonical dimerization interface. Furthermore, their C-terminal F domain forms a short β-sheet with the L8–9 loop holding H12 in an agonistic conformation that is incompatible with the standard dimerization mode (Supplementary Fig. 4a). The current structure of the AR-LBD homodimer now solves the structural dilemma of the quaternary assemblies of oxosteroid receptors by revealing a dimerization mode with a major contribution of H5 from both partners (Supplementary Figs 4 and 5).

Several observations raise the interesting possibility that the AR-like dimeric conformation could be adopted by other members of the oxosteroid subfamily. In this regard, it is noteworthy that residues involved in maintaining the rigid, dimerization-competent structure of H5 such as W752 and F755 as well as F804 (H8) are highly conserved in all oxosteroid receptors. Indeed, the current AR-LBD homodimer shows some resemblance to a previously reported crystal structure of homodimeric GR-LBD[35] (PDB 1M2Z; see also Supplementary Figs 4b and 5a). However, although several topologically equivalent residues contribute to the inter-monomer interface in the 1M2Z structure (Supplementary Fig. 5a), a closer inspection reveals that the 'right side' GR-LBD molecule is

tilted towards the lower half of the dimer (Supplementary Fig. 4b). As a result, the 1M2Z dimer interface is substantially smaller and less intimate: $600\,\text{Å}^2$ buried surface area versus $1,000\,\text{Å}^2$ in the case of the AR homodimer (Fig. 2a,b; Supplementary Fig. 4b). A hypothetical head-to-head model of the GR-LBD homodimer can be straightforwardly generated by a rigid-body rotation of the 'right side' GR-LBD monomer of the previous 1M2Z structure, accompanied by some side chain rotations and repositioning of the highly flexible L9–10 loop to avoid steric clashes across the dimer interface (Supplementary Fig. 4b). Notably for the GR, different physiological functions have been ascribed for receptor monomers versus dimers, but the analyses have been mainly based on DBD-mediated dimerization[26,50]. The herein proposed LBD-mediated dimerization mechanism will need to be taken into consideration in this field as well.

Similarly, our modelling experiments also indicate that head-to-head dimers are perfectly compatible for the PR and MR LBDs. We notice that a topologically unrelated configuration has been reported for the PR-LBD homodimer that includes H12 and is formed upon exclusion of $700\,\text{Å}^2$ of solvent accessible surface[33,34] (Supplementary Fig. 5b). However, this arrangement is more difficult to reconcile with the current models of receptor action because it would occlude the coactivator binding AF-2 pocket. Finally, H5-centred symmetric dimers might be additionally relevant for other NRs. In this regard, previously reported structures of TR homodimers reveal that topologically equivalent elements from H5 and L5–6 contribute to the dimer interface[51,52] (Supplementary Fig. 5c).

In summary, the crystal structure of the AR-LBD homodimer presented here appears to represent a physiologically relevant conformation of the dimeric receptor *in vivo* and allows for a deeper understanding of the mechanisms of action of this important transcription factor. We provide structure-function insights into how pathogenic mutations that cluster at the dimer interface alter key functions of the AR. Furthermore, the current structure suggests an elegant map of allosteric connections between major AR functional sites with important implications for signal transmission across the LBD. How this dimeric structure fits in the context of the AR full-length dimer and its exact roles in spatial-temporal control of gene transcription remain major intellectual challenges with important clinical implications. Even though caution must be exercised when extrapolating the current results to other NRs, the head-to-head structure of the AR-LBD homodimer might represent a common conformation for both homo- and heterodimers of other members of the subfamily of oxosteroid receptors and perhaps for other NRs as well. Moreover, despite the inherent challenges to develop protein–protein disruptors, our structural investigation opens up avenues for therapeutic intervention, as small molecules that may interfere with AR dimerization could be potentially useful for PCa treatment.

## Methods

**Peptides and proteins.** A peptide corresponding to residues $S^{59}$-$V^{72}$ of human UBA3 was custom-synthesized at Pepmic. Recombinant human AR-LBD (residues 662–919) was expressed as a fusion protein with thioredoxin and purified to homogeneity using standard purification methods[23].

**Crystallization and structure elucidation.** Purified, concentrated DHT-bound human AR-LBD (residues 662–919) was combined with a threefold molar excess of synthetic UBA3 $^{59}$S–$^{72}$V peptide and incubated overnight at 4 °C. Drops of the AR-LBD-UBA3 mixture were equilibrated against 0.1 M HEPES, pH 7.5, 1.32 M ammonium sulfate using the sitting drop vapor-diffusion method. Diffraction data were collected at 100 K at ALBA CELLS synchrotron and processed using MOSFLM (http://www.mrc-lmb.cam.ac.uk/harry/mosflm/) and CCP4 (http://www.ccp4.ac.uk/). The crystal structure was solved and refined using MOLREP, REFMAC5 and COOT from the CCP4 package. Crystal packing analysis was performed using PISA (http://www.ebi.ac.uk/), model quality was assessed with MolProbity (http://molprobity.biochem.duke.edu/) and figures were prepared with PyMOL (http://www.pymol.org).

**Surface plasmon resonance analyses.** SPR analyses were performed at 25 °C in a BIAcore T200 instrument (GE Healthcare). Highly purified, DHT-bound recombinant AR-LBD was diluted in 10 mM sodium acetate, pH 4.7 and directly immobilized on a CM5 chip (GE Healthcare) by amine coupling. Two different ligand densities were used: ∼300 resonance units (RU) for AR-LBD self-association experiments, and ≈6,800 RU for studying the binding of the UBA3-derived peptide to AR-LBD. As a reference, one of the channels was also amine-activated and blocked in the absence of protein. Alternatively, in some experiments the unrelated NR, TLX/NR2E1, was coupled in the reference channel. The running buffer was 50 mM HEPES pH 7.2, 50 mM Li$_2$SO4, 5% glycerol, 1 mM DTT, 5 mM DHT. Sensorgrams were analysed with the BIAcore T200 Evaluation software 3.0 and Scrubber2, and fitted according to the 1:1 Langmuir model.

**Cross-linking experiments.** Purified recombinant human AR-LBD (33 µM) was incubated either with glutaraldehyde (GA; 0.05% final concentration; Sigma) for 10 min, or with 200 µM bis(maileimido)ethane (BMOE, Thermo Scientific) or 1-ethyl-3-(3-dimethylaminopropyl)carbodiimide hydrochloride (EDC, Pierce) for 1 h at 37 °C following the manufacturer's instructions. Samples of the reaction mixtures were boiled in the presence of reducing Laemmli sample buffer, and separated by SDS-PAGE.

**SDS-PAGE and western blotting.** Proteins were resolved on 10% SDS-poly-acrylamide gels, which were then either stained with Coomassie Brilliant Blue or electroblotted onto PVDF membranes. Western blot analysis of recombinant AR-LBD was performed using a commercially available anti-AR-LBD antibody (C19; Santa Cruz). For verifying expression levels of full-length AR we used an in-house antibody targeting the N-terminal domain of the protein[53] and antibodies against GAPDH (Santa Cruz) as control. Secondary anti-rabbit (P0217) and anti-mouse antibodies (P0260; both from Dako) conjugated to horseradish peroxidase were used for visualization by immunodetection combined with chemiluminescence (Western Lightning Plus-ECL; PerkinElmer).

**Nanoliquid chromatography tandem mass spectrometry.** Coomassie Brilliant Blue-stained bands corresponding to monomeric and dimeric AR-LBD were excised from the gels and subjected to in-gel digestion following standard protocols[54]. Briefly, excised bands were reduced (10 mM dithiothreitol in 50 mM bicarbonate buffer, pH 8, for 45 min at 56 °C), alkylated (50 mM iodacetamide in 50 mM ammonium bicarbonate buffer for 30 min at 25 °C) and digested with sequencing-grade chymotrypsin (Promega) overnight at 37 °C in 100 mM ammonium acetate buffer, pH 8. Chymotryptic peptides were diluted in 1% formic acid (FA) and loaded onto a 180 µm × 20 mm C18 Symmetry trap column (Waters) at a flow rate of 15 µl min$^{-1}$ using a nanoAcquity Ultra Performance LCTM chromatographic system (Waters). Peptides were separated using a C18 analytical column (BEH130 C18, 75 mm × 25 cm, 1.7 µm; Waters) with a 120-min run, comprising three consecutive linear gradients: from 1 to 35% B in 100 min, from 35 to 50% B in 10 min and from 50 to 85% B in 10 min ($A = 0.1\%$ FA in water, $B = 0.1\%$ FA in CH$_3$CN). The column outlet was directly connected to an Advion TriVersa NanoMate (Advion) fitted on an LTQ-FT Ultra mass spectrometer (Thermo), which was operated in positive mode using the data-dependent acquisition mode. Survey MS scans were acquired in the FT with the resolution (defined at 400 $m/z$) set to 100,000. Up to six of the most intense ions per scan were fragmented and detected in the linear ion trap. The ion count target value was 1,000,000 for the survey scan and 50,000 for the MS/MS scan. Target ions already selected for MS/MS were dynamically excluded for 30 s. Spray voltage in the NanoMate source was set to 1.70 kV. Capillary voltage and tube lens on the LTQ FT were tuned to 40 and 120 V, respectively. The minimum signal required to trigger MS to MS/MS switch was set to 1,000 and activation Q was 0.250. Singly charged precursors were rejected for fragmentation.

**Bioinformatic analysis of the impact of pathogenic mutations.** The following bioinformatics programs were used to assess the impact of point mutations on the stability of monomeric *versus* dimeric AR-LBD conformations: CUPSAT (ref. 55), SMD (ref. 56), Polyphen (ref. 57) and iStable (ref. 58). All mutations were in addition visually inspected using Pymol. Folding RaCe (ref. 59) was used to predict the changes in folding rates upon mutation.

**Cell culture.** PC-3, COS-7 and Hep3B cells were obtained from the American Type Culture Collection (ATCC) and were authenticated by short-tandem repeat DNA profiling by Genetica. PC-3 and COS-7 cells were cultured in Dulbecco's modified Eagle medium supplemented with 10% foetal calf serum, while Hep3B cells were cultured in minimum essential medium (MEM)-α medium with Glu-taMAX supplement and without nucleosides (Gibco).

**Acceptor photobleaching FRET experiments.** To enable detection of AR-LBD dimerization in living cells, a fragment coding for residues 612–919 of the human AR gene was genetically labelled with EYFP (N-terminal) or ECFP (C-terminal). In addition, P767A and Y764C mutations were introduced in these constructs. Acceptor photobleaching (abFRET) was performed following standard protocols[30]. In short, Hep3B cells were seeded on cover glasses and grown in full culture medium (alpha-MEM). Cells were co-transfected with constructs encoding EYFP- and ECFP-labelled AR-LBD. Cells transfected with either free EYFP and ECFP or with the ECFP-EYFP fusion construct served as negative and positive controls, respectively. Transfected cells were grown for at least 16 h in alpha-MEM supplemented with 5% charcoal-stripped serum and (ant-)agonists as described in the figure legends. In abFRET, images of cells expressing both EYFP and ECFP were collected sequentially with a Zeiss LSM510meta confocal microscope. ECFP and EYFP were detected using 514 and 458 nm excitation at moderate laser power, and the emission was detected using a 470–500 nm bandpass emission filter and a 560 nm-long pass emission filter, respectively. After image collection, EYFP was bleached in the whole nucleus by scanning a region of 200 µm$^2$ 25 times at 514 nm at high laser power. After photobleaching, a second EYFP and ECFP image pair was collected. Apparent FRET efficiency was calculated using the equation:

$$\mathrm{abFRET} = [(\mathrm{ECFP}_{after} - \mathrm{ECFP}_{before}) \times \mathrm{EYFP}_{before}] \big/ [(\mathrm{ECFP}_{after} \times \mathrm{EYFP}_{before}) - (\mathrm{ECFP}_{before} \times \mathrm{EYFP}_{after})], \quad (1)$$

where $\mathrm{ECFP}_{before}$ and $\mathrm{EYFP}_{before}$ are the mean prebleach fluorescence intensities of ECFP and EYFP, respectively, in the area to be bleached (after subtraction of background), and $\mathrm{ECFP}_{after}$ and $\mathrm{EYFP}_{after}$ are their mean postbleach fluorescence intensities, in the bleached area. The apparent FRET efficiency was finally expressed relative to control measurements in cells expressing either free ECFP and EYFP ($\mathrm{abFRET}_0$) or the ECFP-EYFP fusion protein ($\mathrm{abFRET}_{\mathrm{ECFP\text{-}EYFP\ fusion}}$):

$$\mathrm{apparent\,FRET\,efficiency} = (\mathrm{abFRET} - \mathrm{abFRET}_0) / (\mathrm{abFRET}_{\mathrm{ECFP\text{-}EYFP\ fusion}} - \mathrm{abFRET}_0). \quad (2)$$

**Transactivation assays.** PC-3 cells were seeded in 96-well plates (10,000 cells per well) and transfected with 100 ng of reporter construct, 10 ng of AR expression vector and 5 ng of pCMV-β-gal expression vector. Hormone treatments were done in triplicate; transfections were done in biological quadruplicates. After transfection, the cells were stimulated with increasing concentrations of DHT. The following day, cells were harvested in Passive Lysis buffer and luciferase as well as β-galactosidase activity were measured with a Luminoskan luminometer[53].

**Whole-cell competition assay.** COS-7 cells were seeded in 48-well plates at a density of 30,000 cells per well and transfected with 375 ng of AR and 75 ng of pCMV-β-gal expression vectors, respectively. On the next day, cells were incubated with 1 nM [$^3$H]-labelled mibolerone and increasing concentrations of unlabelled DHT (0.1 nM–10 μM). After incubation at 37 °C for 90 min, cells were washed three times with ice cold PBS and lysed in Passive Lysis buffer. Radioactivity present in these extracts was determined by liquid scintillation counting.

**Ligand-binding assay.** COS-7 cells were seeded in 48-well plates at a density of 30,000 cells per well and transfected with 375 ng of AR and 75 ng of pCMV-β-gal expression vectors, respectively. On the next day, cells were incubated with a range of concentrations of [$^3$H]-labelled mibolerone or with the same concentrations of the unlabelled ligand. After incubation at 37 °C for 90 min, cells were washed three times with ice cold PBS and lysed in Passive Lysis buffer. Radioactivity present in these extracts was determined by liquid scintillation counting. Maximal binding was determined by incubating the cells with 300 nM labelled mibolerone.

**Data availability.** The atomic coordinates and structure factors have been deposited in the PDB (www.rcsb.org) and the accession code assigned is 5JJM. The PDB accessibility has been designed 'for immediate release on publication'. The following PDB accession codes were used in this work: 1I38, 1T76, 1T63, 1XQ3, 1T5Z, 1Z95, 1XNN, 2IHQ, 2HVC, 2PIX, 2PIU, 2PIR, 2PIO, 3L3Z, 4OH5, 1M2Z, 1A28, 3D57, 4IQR, 1DKF and 3E00. The proteomic data sets have been deposited in the PRIDE repository and the data set is available via ProteomeXchange with the data set identifier PXD005575. The authors declare that all data supporting the findings of this study are available within the article and its Supplementary Information Files, or available from the authors on reasonable request.

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

## Acknowledgements

E.E.-P. acknowledges the kind generosity and support of Gemma E. Carretero Fund for Science. We thank Arnold T. Hagler, Robert J. Fletterick, Robert Huber, Joseph Schlessinger, Jens Lüders, David T. Mangelsdorf and further members of the NR community for stimulating discussions. We thank Jordi Juanhuix and coworkers for assistance at the ALBA-Xaloc beamline. The grants FPDI-2013–18489, SAF2014-57994-R, SAF2015–71878 (MINECO, Gobierno de España), the Serra Húnter Programme and 2014-SGR-01214 (Generalitat de Catalunya) are acknowledged. F.C. is holder of a grant from FWO-Vlaanderen (G.0684.12N, G.0830.13N), from the KU Leuven (GOA/15/017) and a grant from 'Kom op tegen kanker'. The IRB-Mass Spectrometry Core Facility participates in the BMBS European Cost Action BM 1403, Proteored (PRB2 IPT13/0001) and the Severo Ochoa Award (MINECO, Gobierno de España).

## Author contributions

Conception and design by F.C., P.F-P., E.E.-P.; Data collection by M.N., S.P., N.G., C.H., M.A., K.Z., M.G., M.T., M.E.R. and E.E-P; Analysis and interpretation by M.N., S.P., N.G., C.H., M.G., M.V., M.T., A.H., M.E.R., F.C., P.F-P. and E.E-P; Writing the article by F.C., P.F-P., E.E-P. Critical revision of the article by A.H., M.E.R., F.C., P.F-P., E.E-P.; Final approval of the article by A.H., M.E.R., F.C., P.F-P., E.E-P.; Funding obtained by N.G., F.C., P.F-P., E.E-P.; Overall responsibility by P.F-P., E.E-P.

## Additional information

**Competing financial interests:** The authors declare no competing financial interests.

