## [Peer Review File · Nature Communications]

Reviewer #1 (Remarks to the Author)

The manuscript by Gallastegui et al. reports critical new findings based on the crystal structure of the androgen receptor (AR) ligand binding domain (LBD) homodimer. The report first shows that AR LBD forms a previously unknown mode of subunit dimerization, this is much bigger in surface area than those seen in PR and GR, and the self-association interface consists of a symmetric organization mode. The amino-acid residues in this interface are particularly susceptible to mutations in human disease settings, and the manuscript does an extraordinary job at clarifying how these mutations at the dimer interface disrupt the interface. The entire work is nicely supported by a substantial battery of quantitative biochemical and functional studies. Together, all the data allow the authors to firmly establish the importance of their novel structural findings and the physiological importance of the dimer interface. The crystal structure itself is also high resolution (2.15 Å) and has been carefully modeled and analyzed. The text of the manuscript is clearly written and easy to follow for general readers from the cancer field, in addition to those already knowledgeable about nuclear receptor architectures.

Overall, I can strongly support its publication in Nature Communications, and believe its publication is merited given the impressive scientific quality, the novelty of the findings and the fundamental importance of these findings to both the androgen receptor and prostate cancer fields. There is indeed much in this manuscript that reshapes our past thinking about steroid receptor LBD organization, and a great amount of validation is provided here too.

However, I have a few notes for improving the text, written below:

1. On the bottom of page 3, the statement and reference reading: "*small-angle X-ray scattering, or electron microscopy (Rochel et al., 2011; Orlov et al., 2012)*" . The authors should make a note here that the small-angle x-ray studies reference by Rochel et al. provided poorly resolved models and still lack evidence for the true presence of DNA in these complexes. their findings have been challenged subsequently. A reference to Nuclear receptor full-length architectures: confronting myth and illusion with high resolution, by Rastinejad F, Ollendorff V, Polikarpov I. Trends Biochem Sci. 2015 Jan;40(1):16-24 should follow here to point readers to potentially erroneous conclusions made by Rochel et al.. Additionally, another reference is needed too, as the Polikarpov lab has carried out solution studies on the PPAR-RXR-DNA bound and DNA-free full-length complexes that deserve to be referenced when discussing full-length receptor structures. These findings have also contradicted the Rochel conclusions based on SAXS.
2. On page 11, the sentence "Thus, there is evidence for inter- and intramolecular interactions at the levels of the N/C interactions (Langley et al., 1995), of the DBD (Shaffer et al., 2004), and of the LBD (this work). The word "Thus" should be taken out, as it adds some confusion as it would seem to suggest that the N/C interaction information comes from the current findings (mentioned in the previous sentence).
3. On page 12, "Several of these AIS-associated mutations lead to local structural disorder of the AR monomer, which explains their pathogenicity". I think the authors should more correctly state "may lead" rather than "lead" in this sentence, since there is no direct evidence provided in their work that some of these mutations actually cause local structural disorder.
4. On page 14, additional remarks are needed within the section "allosteric communication across the ligand-binding domain". The authors should clearly clarify and distinguish their type of allosteric communication here from the more controversial allosteric communication proposed by the Moras group, also between LBDs of homodimers.

To summarize the issue: in a PNAS paper (J. Osz et al. 2012, Moras group) used RAR homodimers and concluded that the coactivator peptide binding to one LBD allosterically precludes the binding

of a second coactivator peptide to the remaining LBD of receptor homodimers. The current crystal structure here has both LBDs showing equivalently bound coactivator peptides, which fails to support the allostery indicated by Osz et al. It would be appropriate here to also cite the valuable work of P. Yi and co-authors from the B O'Malley lab ("Structure of a biologically active estrogen receptor-coactivator complex on DNA, published in 2015", which shows that two full-length SCR coactivators do in fact bind at the same time to the two LBDs of full-length ER. Based on their EM studies, the O'malley paper concludes that the LBD-LBD allostery suggested by the Moras group to also form in steroid receptor LBDs failed to see validation within the structural complex of full-length receptors (ER) and SRC proteins that were studied by EM. All of these related issues around allostery truly deserve some discussion in this section.

5. The last statement of the main text: "our structural investigation opens up novel therapeutic avenues for inhibiting the AR". It is not obvious how the findings here could directly open up new therapeutic avenues; perhaps the authors can add a sentence or two more to solidify this suggestion, or remove it.

Minor points:

A period is missing on page 15, bottom paragraph between "residues" and "It, among the words : "LBD residues) It is no.." And another period is missing in the penultimate sentence on page 17 before the word "Moreover".

Reviewer #2 (Remarks to the Author)

This manuscript describes a valuable new contribution to the scientific community with an expected broad audience interested in the experiments done and findings described in the manuscript.

The editor requested me to look at the SPR and MS experiments and data. Therefore, I focused only on the SPR and MS experiments, which are my expertise. Overall, I conclude that the experiments are performed the way they should be. I have some questions and suggestions to strengthen the conclusions drawn from the experiments performed:

-Figure 3a: From the concentrations tested in the K_d determination with SPR, it appears that a serial dilution is used in which 1:1 dilution was performed. Why is the measurement between 3.75 and 0.937 not shown, or not performed? Logically, the concentration of 1.875 μ M should be there too?

-Figure 3a: Why was the RU response not measured until saturation binding?

-SPR: Why was "no protein immobilized" used as reference channel? Using another NR-LBD (for example the ER-LBD) or other protein would have been a valuable additional negative control to also monitor non-specific protein binding of the AR-LBD.

-Figure 3b: What exact curve fitting (parameters used for the curve fit) was used to calculate the K_d and why? It looks like the curve fitting used does not allow the curve to go through the points at the lower slowly increasing part of the curve. With variable slope fitting, I expect a better fit.

-Why was the 1:1 Langmuir model used for fitting?

-To further strengthen the SPR data I suggest an additional experiment with antagonist-bound AR-LBD as negative control.

-I suggest to also do an MS crosslinking experiment on antagonist bound AR-LBD as additional negative control.

Reviewer #3 (Remarks to the Author)

This paper reports the structure of the Androgen Receptor ligand binding domain (ARLBD) bound to the agonist DHT and a peptide that binds in a surface groove like established co-activator proteins.

Given that there are other structures of the ARLBD bound to DHT and co-activators, the novelty of this paper lies in the fact that two of the four copies of the ARLBD in the asymmetric unit appear to form a homodimer and that residues at the homodimeric interface have been found to play an important role in AR mediated disease. This is potentially important since it is believed that the full length receptor functions as a dimer, but as yet the mode of dimerization is unclear. The DBDs dimerize in DNA, but it is presumed, but not shown, that the LBDs might also dimerize.

It is interesting that the proposed dimerization interface is the location of a number of deleterious mutations. In general one would only expect surface mutations to have a strong phenotype if the surface were involved in mediating important intermolecular interactions. These could of course be with with another protein or indeed homodimerization as proposed here.

I remain to be fully convinced that the interface observed is a bona fide homodimerization interface and in my view it is essential that this is proven. After all, why do all the other structures not show this dimer interface? and why do two of the four molecules in this crystal also not show this dimer interface?

The buried surface area is reasonably large, but nevertheless could still be a crystal packing interface. The glutaraldehyde cross linking experiments show oligomerization and so are not helpful. The BMOE cross linker only cross-links cysteines and there may only be one suitable cysteine that can be cross linked - hence only producing dimers.

I think more direct biophysical techniques are required - such as analytical ultracentrifugation or multi angle light scattering after size exclusion chromatography.

To fully support the arguments in this paper, it would be essential to show that the dimer is disrupted by the mutations at the proposed dimer interface.

Finally, the discussion is far too long and unbalances the paper. Much could go into supplementary material.

Minor point: The temperature factor comparison in Sup Fig 1 only makes sense if the scale is absolute (not simply relative within each PDB file). This is not clear from the description

DETAILED RESPONSES TO THE REVIEWERS

Our first submission was well received, but the reviewers made specific requests that we have addressed by making the following changes and additional experiments as detailed below:

Reviewer 1:

1. On the bottom of page 3, the statement and reference reading:

", small-angle X-ray scattering, or electron microscopy (Rochel et al., 2011; Orlov et al., 2012)". The authors should make a note here that the small-angle x-ray studies reference by Rochel *et al.* provided poorly resolved models and still lack evidence for the true presence of DNA in these complexes. Their findings have been challenged subsequently. A reference to Nuclear receptor full-length architectures: confronting myth and illusion with high resolution, by Rastinejad F, Ollendorff V, Polikarpov I. Trends Biochem Sci. 2015 Jan;40(1):16-24 should follow here to point readers to potentially erroneous conclusions made by Rochel *et al.*. Additionally, another reference is needed too, as the Polikarpov lab has carried out solution studies on the PPAR-RXR-DNA bound and DNA-free full-length complexes that deserve to be referenced when discussing full-length receptor structures. These findings have also contradicted the Rochel conclusions based on SAXS.

In response to this comment, we have rephrased the sentence:

"Structural information has been gathered on several full-length NRs either by X-ray crystallography (Chandra et al., 2008; Chandra et al., 2013; Lou et al., 2014), small-angle X-ray scattering, or electron microscopy (Rochel et al., 2011; Orlov et al., 2012)",

which now reads as follows:

"Structural information has been gathered on several full-length NRs either by **detailed X-ray crystallography** (Chandra et al., 2008; Chandra et al., 2013; Lou et al., 2014), or through small-angle X-ray scattering and electron microscopy **at lower resolution** (Rochel et al., 2011; Orlov et al., 2012; **Bernades et al., 2012; Belorusova et al., 2016; Yi et al., 2015**)".

(Note that we refer to the recently published articles by Yi and co-workers (2015) and by Belorusova and co-workers (2016) in the revised version of the manuscript, as requested). Furthermore, and following this comment, we have added the following sentence:

"**These biophysical investigations have revealed conflicting data that does not allow a unified paradigm of full-length NR architecture at present time (Rastinejad et al., 2015; Moras et al., 2015)**".

Unfortunately, a detailed discussion of the interesting controversy regarding NR architecture is out of the scope of the current manuscript.

2. On page 11, the sentence "Thus, there is evidence for inter- and intramolecular interactions at the levels of the N/C interactions (Langley et al., 1995), of the DBD (Shaffer et al., 2004), and of the LBD (this

work). The word "Thus" should be taken out, as it adds some confusion as it would seem to suggest that the N/C interaction information comes from the current findings (mentioned in the previous sentence).

To avoid misinterpretations, the word "Thus" has been removed in the revised version of the manuscript. The sentence is now on page 12.

3. On page 12, "Several of these AIS-associated mutations lead to local structural disorder of the AR monomer, which explains their pathogenicity". I think the authors should more correctly state "may lead" rather than "lead" in this sentence, since there is no direct evidence provided in their work that some of these mutations actually cause local structural disorder.

"Lead" has been changed to "may lead" in this sentence, which appears as part of the Supplementary Information in the revised version of the manuscript.

4. On page 14, additional remarks are needed within the section "allosteric communication across the ligand-binding domain". The authors should clearly clarify and distinguish their type of allosteric communication here from the more controversial allosteric communication proposed by the Moras group, also between LBDs of homodimers.

To summarize the issue: in a PNAS paper (J. Osz et al. 2012, Moras group) used RAR homodimers and concluded that the coactivator peptide binding to one LBD allosterically precludes the binding of a second coactivator peptide to the remaining LBD of receptor homodimers. The current crystal structure here has both LBDs showing equivalently bound coactivator peptides, which fails to support the allostery indicated by Osz et al. It would be appropriate here to also cite the valuable work of P. Yi and co-authors from the B O'Malley lab ("Structure of a biologically active estrogen receptor-coactivator complex on DNA, published in 2015", which shows that two full-length SCR coactivators do in fact bind at the same time to the two LBDs of full-length ER. Based on their EM studies, the O'Malley paper concludes that the LBD-LBD allostery suggested by the Moras group to also form in steroid receptor LBDs failed to see validation within the structural complex of full-length receptors (ER) and SRC proteins that were studied by EM. All of these related issues around allostery truly deserve some discussion in this section.

We agree with the reviewer, and in the revised version of the manuscript we now cite the paper by O'Malley's group, both in the Introduction (page 3):

"Structural information has been gathered on several full-length NRs either by detailed X-ray crystallography (Chandra et al., 2008; Chandra et al., 2013; Lou et al., 2014), or through small-angle X-ray scattering and electron microscopy at lower resolution (Rochel et al., 2011; Orlov et al., 2012; Bernades et al., 2012; Belorusova et al., 2016; **Yi et al., 2015**)",

and in the Discussion (page 15):

"Of note, the AF-2 pockets of both partners in the AR dimer remain accessible for interactions with coregulatory complexes, **in alignment with currently accepted models of full-length NR functioning derived from EM data** (Yi et al., 2015)".

5. The last statement of the main text: "our structural investigation opens up novel therapeutic avenues for inhibiting the AR". It is not obvious how the findings here could directly open up new therapeutic avenues; perhaps the authors can add a sentence or two more to solidify this suggestion, or remove it.

In the revised version of the manuscript we have added a final sentence to stress our belief in the druggability of the newly unveiled molecular surface, although admitting that development of molecules that target the homodimer interface might not be straightforward:

“Moreover, despite the inherent challenges to develop protein-protein disruptors, our structural investigation opens up avenues for therapeutic intervention, as small molecules capable of interfering with AR dimerization could be potentially useful for PCa treatment”.

6. Minor points:

A period is missing on page 15, bottom paragraph between "residues" and "It, among the words : "LBD residues) It is no..". And another period is missing in the penultimate sentence on page 17 before the word "Moreover".

These typos have been corrected in the revised version of the manuscript.

Reviewer 2:

1. Figure 3a: From the concentrations tested in the K_d determination with SPR, it appears that a serial dilution is used in which 1:1 dilution was performed. Why is the measurement between 3.75 and 0.937 not shown, or not performed? Logically, the concentration of 1.875 μM should be there too?

The measurements at the intermediate concentration (1.875 μM) had been performed, but were not included in the original Fig. 3a by mistake. We have now included the re-calculated sensorgrams in the new Fig. 3a, and recalculated the kinetics and affinity constants (page 7 in the revised version of the manuscript).

2. Figure 3a: Why was the RU response not measured until saturation binding?

We did perform measurements up to an AR-LBD concentration of 30 μM, but these data were not included in the original Fig. 3 because saturation was already achieved at a lower concentration (15 μM). We have now included all data up to the 30 μM concentration in both sensorgrams, kinetics and affinity, in the new Fig. 3, panels a and b, respectively.

3. SPR: Why was "no protein immobilized" used as reference channel? Using another NR-LBD (for example the ER-LBD) or other protein would have been a valuable additional negative control to also monitor non-specific protein binding of the AR-LBD.

Our SPR expert suggests to activate the reference channel surface with NHS/EDC and then deactivate (block) it with ethanolamine without immobilizing a reference protein when the molecular masses of protein ligands are below 150 kDa. Nevertheless, we have performed additional SPR experiments with the ligand binding domain of the orphan receptor TLX (NR2E1) as suggested by the reviewer. Essentially the same results were obtained irrespective of whether the reference channel was left unoccupied or using TLX-LBD.

4. Figure 3b: What exact curve fitting (parameters used for the curve fit) was used to calculate the K_d and why? It looks like the curve fitting used does not allow the curve to go through the points at the lower slowly increasing part of the curve. With variable slope fitting, I expect a better fit.

We have applied global fitting for the steady state plot (1:1) and the offset adjusted to zero.

5. Why was the 1:1 Langmuir model used for fitting?

According to current models of nuclear receptor functioning supported by strong biophysical evidence, LBD dimers are the functional biological unit. Therefore, the 1:1 Langmuir model was chosen for fitting.

6. To further strengthen the SPR data I suggest an additional experiment with antagonist-bound AR-LBD as negative control. I suggest to also do an MS crosslinking experiment on antagonist bound AR-LBD as additional negative control.

Despite our repeated attempts over the years to recombinantly express human wild-type AR-LBD in *E. coli* in the presence of antiandrogens (enzalutamide, flutamide and bicalutamide, among others), we have not been able to obtain enough soluble protein for further functional and biophysical studies. So, these negative controls cannot be performed at this point. However, our FRET experiments provide direct evidence that DHT and other AR agonists induce AR-LBD homodimerization, while all tested antiandrogens failed to show any effect in this regard.

Reviewer 3:

1. I remain to be fully convinced that the interface observed is a bona fide homodimerization interface and in my view it is essential that this is proven.

In the light of this criticism, we have performed new experiments in cells by means of FRET to further substantiate our findings. We could demonstrate that the human AR-LBD dimerizes in the crowded cellular environment in response to DHT and other AR agonists, but not in the presence of clinically used antiandrogens. Of particular note in the light of the reviewer's criticism, a CAIS-linked mutation, P767A, impaired homodimer formation, as indicated by the absence of FRET signal. Most importantly, this blockade of dimerization was not due to a major impact on protein folding, as residue P767 is solvent exposed in the LBD monomer, and the P767→A mutant still binds androgens, both when present in an isolated LBD (Fig. 4d) or in full-length AR (Fig. 5b,c). These new experiments, added to the structural and functional evidence already discussed in the previous version of our work, provides compelling evidence for the (patho)physiological relevance of the AR-LBD dimerization mechanism presented for the first time in the accompanying manuscript.

2. After all, why do all the other structures not show this dimer interface? and why do two of the four molecules in this crystal also not show this dimer interface? The buried surface area is reasonably large, but nevertheless could still be a crystal packing interface.

As we discuss in our manuscript, the previously reported crystal structures of homodimeric GR-LBD and TR-LBD suggest that the dimerization mechanism unveiled in the current structure might be shared by other nuclear receptors as well, perhaps with some variations. Regarding why the homodimeric conformation of the AR-LBD has not been observed before, of course, one can only speculate. As the reviewer is probably aware of, during crystal formation there is always a competition between the intermolecular forces that stabilize the multimeric arrangement(s) of a particular protein, on the one hand, and those imposed by favourable crystal contacts, on the other. As the result of these competing forces, different outcomes are possible (and are well documented in protein crystallography): from multimeric proteins that crystallize following the symmetry of the crystal lattice (e.g., a physiologically relevant dimer in solution could crystallize with one monomer in the ASU, while the second monomer is related to this one by an exact two-fold axis), to cases in which "crystal dimers" are mere artefacts. In our case, we believe that the results of our FRET experiments (Fig. 4) and of the functional studies (Fig. 5), in addition to the crosslinking experiments with BMOE *in vitro* (Fig. 3) provide strong support for our interpretation that the surface centered on helix H5 corresponds to the *bona fide* homodimerization interface. Probably, the presence of a divalent cation bound to residue C853 (Supplementary Fig. 2), might have helped to organize the L9-10 loop, and thus tipped the balance towards the dimeric form.

A final word on this issue: not only is the buried surface area "reasonably large" (1,000 Å²), but its biological relevance is also supported by the overall evolutionary conservation of

interface residues and by the results of algorithms that differentiate between biologically meaningful assemblies from crystal artefacts. For instance, protein-protein interaction sites are known to be enriched in a restricted set of specialized interface residues (Pro, Ile, Tyr, Trp, Asp and Arg; ref. Sillerud and Larson, 2005), in excellent agreement with those identified in the present crystal structure.

3. The glutaraldehyde cross linking experiments show oligomerization and so are not helpful. The BMOE cross linker only cross-links cysteines and there may only be one suitable cysteine that can be crosslinked - hence only producing dimers.

We agree that our preliminary glutaraldehyde crosslinking experiments do not allow to discriminate between the dimerization interface unveiled in our crystal structure and other possible arrangements. This is precisely the reason why BMOE was chosen for crosslinking experiments: the Cys687– Cys687' pair is the only one in which two side chain SH groups from neighboring monomers are located close enough to be connected by a BMOE arm. (See the schematic representation of the distribution of cysteine residues in the AR-LBD homodimer below). To help the readers follow our reasoning, we have now included the following sentence:

“First, we notice that four out of the six cysteine residues in the AR-LBD are solvent accessible (C670, C687, C845 and C853), and thus capable of reacting with sulfhydryl-reactive small molecules. Further, inspection of the homodimer structure immediately reveals that only residues C687 from the two core monomers are located close enough to be simultaneously engaged by the short-arm crosslinker, bis-maleimidoethane (BMOE) (Fig. 3c)“. (Pages 7-8 in the revised version of the manuscript).

PISA calculations: <http://www.ebi.ac.uk/pdbe/pisa/>

C670: solvent accessible

C687: interfacing residue

C785: Inaccessible

C807: Inaccessible

C845: interfacing residue

C853: solvent accessible

Legend: Three-dimensional structure of the AR-LBD homodimer, indicating the positions of all cysteine residues. The estimated surface accessibility of these cysteines is indicated (as assessed with PISA).

4. I think more direct biophysical techniques are required - such as analytical ultracentrifugation or multi angle light scattering after size exclusion chromatography.

We did consider performing further biophysical experiments in addition to those presented in the previous version of our manuscript, but they all suffer from the same limitation: they do not allow distinguishing between different dimer conformations. More importantly, the results of our new FRET experiments have allowed us to verify dimer formation in a cellular setting, which as expected is exclusively triggered by AR agonists.

5. To fully support the arguments in this paper, it would be essential to show that the dimer is disrupted by the mutations at the proposed dimer interface.

We have directly addressed this issue by performing FRET experiments, as explained in detail in the revised version of the manuscript (see also the new Fig. 4). Briefly, we have demonstrated that the CAIS-associated mutation, P767A, does not affect ligand binding but fully disrupts dimer formation in a cellular setting. The results of these experiments corroborate that hormone binding and homodimerization are separate but intimately intertwined processes.

6. Finally, the discussion is far too long and unbalances the paper. Much could go into supplementary material.

In response to this criticism, the Discussion section has been shortened. We have in particular moved the detailed analysis of interface mutations identified in individuals with AIS or PCa to the Supplementary Results section.

7. Minor point: The temperature factor comparison in Sup Fig 1 only makes sense if the scale is absolute (not simply relative within each PDB file). This is not clear from the description.

Even though the representations are based on the reported B factors, the fact that all deposited crystal structures of the AR-LBD refined with data in the 1.6-2.4 Å resolution range show significantly higher flexibility than the current homodimer supports the notion that dimerization is associated with an overall stabilization of the domain structure, at least qualitatively. Along these lines, we have observed an overall decrease in stability in mutants of interface residues produced in *E. coli*, as demonstrated by their higher protease sensitivity, marked tendency to self-aggregation, and co-purification with chaperones (data not shown).

Reviewer #1 (Remarks to the Author)

The authors have carefully revised the manuscript to reflect the suggestions I made regarding adding new references and further explanations for some points. I continue to believe this is an important contribution and deserves publication.

Reviewer #2 (Remarks to the Author)

The authors have adequately addressed my suggestions and questions. The manuscript should be ready for publication in my opinion. (Only the MS experiments could not be done with antagonist bound AR LBD. I can in this case suggest buying the AR LBD from Thermofisher for example, but I don't know the implications on the His-tag and/or GCT tag... This is merely a suggestion).

Reviewer #3 (Remarks to the Author)

I still have significant concerns....

Points 1, 4, 5.

I suggested that more direct biophysical techniques were required as well as mutation of the proposed dimerization interface. These points have not been addressed. The authors have presented FRET experiments instead.

At face value, the FRET experiments suggest that when the proteins are overexpressed in the cell, the LBDs will homodimerize in the presence of agonist. The FRET efficiency of 0.1 (10%) seems rather low for a direct interaction. Given that this is such a crucial experiment, it is important that raw data are presented including images of cells post-FRET with and without agonist (also WT versus mutant).

Additionally, in figure 4c the authors do not say which hormone they are using in the FRET assay. What is the reason for switching to mibolerone in figure 4d and 4e?

Point 7.

The way in which the B-factors have been compared in supplementary figure 1 is not valid. This point has NOT been addressed. Pymol scales B-factors from lowest to highest from blue to red. Hence if you have two files with a different range of B-factors then the same value for the B-factor will be colored differently. A single high B-factor water molecule will totally change the color range. In this figure the structures are not colored using the same scale. Therefore, this figure and the arguments based on it, are erroneous and must be removed.

Indeed, the average B-factor in this new structure is 40.54 (supplementary Table 1). The average B-factor in the published structure 1i38 (first structure Supplementary figure 1 panel b) is 34.5 (doi 10.1073/pnas.081565498). How does this prove that the new structure is less flexible?

In supplementary table 1 the authors should report the average B-factor for the solvent, ligand and protein separately.

Reviewer #4 (Remarks to the Author)

The authors measured donor fluorescence after receptor photobleaching, and this is very straightforward way to assess if FRET occurs or not. Although this method will have serious shortcomings for more advanced applications, it does give a + or - readout that is reliable. In this case the authors ask if FRET occurs or not, and find that in some cases it does, and in some cases it does not. The interpretation of the data, in terms of + or -, seems straightforward.

DETAILED RESPONSES TO THE REVIEWERS

Our previous submissions were well received, and 3 out of 4 of the Reviewers are in favour of publication as well as the Editors. Reviewer #3 has made specific requests that we address in the following lines detailed below:

Reviewer #1:

1. Reviewer #1 has accepted all the manuscript changes we have carefully made and agrees with the publication of our manuscript.

Reviewer #2:

1. Reviewer #2 has accepted all the manuscript changes and the Biacore / SPR additions we have introduced and agrees with the publication of our manuscript.

2. Reviewer #2 has suggested the authors to purchase AR-LBD from a commercial source to be able to bypass the problems encountered with the inability to express and purify recombinantly AR-LBD in the presence of antiandrogens. In this regard, we would like to state that we have tested AR-LBD from different commercial sources, but none of the samples purchased were of the quality required to perform reliable experiments. We do stress one more time that the FRET results *in vivo* unambiguously demonstrate that AR-LBD dimerization is impaired in the presence of antiandrogens.

Reviewer #3:

1. "I suggested that more direct biophysical techniques were required as well as mutation of the proposed dimerization interface. These points have not been addressed. The authors have presented FRET experiments instead."

As mentioned in our previous response, we did consider performing further biophysical experiments in addition to those presented in the first version of our manuscript, but they all suffer from the same limitation: they do not allow distinguishing between different dimer conformations. In this regard, the results of the crosslinking experiments with BMOE, verified by MS, unambiguously and directly demonstrated that the dimeric conformation observed in our crystals is also preferred in solution. More importantly, the results of our FRET experiments have allowed us to verify dimer formation in a cellular setting, which as expected is exclusively triggered by AR agonists.

2. "At face value, the FRET experiments suggest that when the proteins are overexpressed in the cell, the LBDs will homodimerize in the presence of agonist."

To avoid artifacts induced by too high (supraphysiological) protein concentrations, for FRET assays we picked cells that have relatively low fluorescence intensities, corresponding to physiological expression

levels. The Houtsmuller and van Royen groups have ample experience with this technique and in its use for studying protein-protein interactions inside living cells (see for example Supplemental Figure S1 in van Royen et al., *J. Cell Biol.*, 2007). From these and other earlier experiments we can estimate expression levels on the basis of the fluorescence intensities (frequently measured on a routine base in our imaging centre) and the microscope settings used.

3. “The FRET efficiency of 0.1 (10%) seems rather low for a direct interaction.”

We used cells separately expressing CFP and YFP molecules, on the one hand, and cells expressing fused CFP-YFP, on the other, as 0 and 100% FRET controls, respectively. The FRET efficiencies we report are calculated relative to these controls. In previous work (see e.g. van Royen et al., *J. Cell Sci.* 2010) we have reported relative FRET efficiencies of cells expressing wild-type AR variants tagged with either CFP or YFP that are in the same range as reported here for the AR-LBD constructs (~10%).

4. “Given that this is such a crucial experiment, it is important that raw data are presented including images of cells post-FRET with and without agonist (also WT versus mutant).”

We are surely willing to provide this data to the Reviewer #3 (pre- and postbleach intensities of all conditions in which LBDs were studied as well as of the CFP, YFP and CFP-YFP fusions). With regard to the request for post-FRET images, we emphasize that the subtle intensity differences in pre- and postbleach images (which are reproducible and statistically significant), can hardly or not be seen at all by naked eye. However, we are willing to provide the requested images to the referee. We will also provide these files upon request to any other scientist interested in reproducing our results, as stated in the Data Availability Statement.

5. “Additionally, in Figure 4c the authors do not say which hormone they are using in the FRET assay. What is the reason for switching to mibolerone in Figures 4d and 4e?”

We have used mibolerone in our studies because it has traditionally been the oral anabolic steroid of choice in nuclear receptor studies.

6. “The way in which the B-factors have been compared in supplementary figure 1 is not valid. This point has NOT been addressed. Pymol scales B-factors from lowest to highest from blue to red. Hence if you have two files with a different range of B-factors then the same value for the B-factor will be colored differently. A single high B-factor water molecule will totally change the color range. In this figure the structures are not colored using the same scale. Therefore, this figure and the arguments based on it, are erroneous and must be removed. Indeed, the average B-factor in this new structure is 40.54 (supplementary Table 1). The average B-factor in the published structure 1i38 (first structure Supplementary figure 1 panel b) is 34.5 (doi 10.1073/pnas.081565498). How does this prove that the new structure is less flexible?”

First of all, we would like to point out to the reviewer that the files used for the qualitative representations of B-factors only included residues, not solvent molecules. Thus, “a single high-B water molecule” will not change anything in these figures. That said, it is true that a rigorous comparison of B-factors between our current dimeric form and all previously deposited structures of monomeric AR-LBD would require re-evaluation of raw diffraction data and crystallographic refinement using exactly the same parameters, including the same set of reflections used for calculating free R factor and so on. This was not our aim, but to provide the readers with representations based on the reported B-factors of all deposited crystal structures of the AR-LBD refined with data in the 1.6-2.4 Å resolution range. We stress that these structures correspond to recombinant molecules produced in different laboratories, crystallized in the presence of different coregulators and agonists or antagonists, measured using different synchrotron sources, and finally refined using different software and/or parameters. The fact that all of these monomeric structures show significantly higher flexibility than the current homodimer supports the notion that dimerization is associated with an overall stabilization of the domain structure, at least qualitatively. Along these lines, we have observed an overall decrease in stability in mutants of interface residues produced in *E. coli*, as demonstrated by their higher protease sensitivity, marked tendency to self-aggregation, and co-purification with chaperones (data not shown).

Reviewer #4:

1. Reviewer #4 has found our FRET experiments properly conducted, and data appropriately interpreted.